# UNDERSTANDING LAYER SIGNIFICANCE IN LLM ALIGNMENT

## ABSTRACT

Aligning large language models (LLMs) through fine-tuning is essential for tailoring them to specific applications. Therefore, understanding what LLMs learn during the alignment process is crucial. Recent studies suggest that alignment primarily adjusts a model's presentation style rather than its foundational knowledge, indicating that only certain components of the model are significantly impacted. To delve deeper into LLM alignment, we propose to identify which layers within LLMs are most critical to the alignment process, thereby uncovering how alignment influences model behavior at a granular level. We propose a novel approach to identify the important layers for LLM alignment (ILA). It involves learning a binary mask for each incremental weight matrix in the LoRA algorithm, indicating the significance of each layer. ILA consistently identifies important layers across various alignment datasets, with nearly 90% overlap even with substantial dataset differences, highlighting fundamental patterns in LLM alignment. Experimental results indicate that freezing non-essential layers improves overall model performance, while selectively tuning the most critical layers significantly enhances fine-tuning efficiency with minimal performance loss.

## 1 INTRODUCTION

Aligning large language models (LLMs) with specific requirements is essential for enhancing their utility across diverse applications (Luo et al., 2023a; Yu et al., 2023; Luo et al., 2023b; Li et al., 2023). Fine-tuning LLMs during the alignment process can significantly improve the models' capabilities to meet targeted needs (Bubeck et al., 2023). Typically, alignment involves fine-tuning the model on diverse datasets, which may include both human-curated (Rajani et al., 2023) and LLM-generated (Taori et al., 2023) data. Such fine-tuning approaches encompass instruction tuning (Wei et al., 2021) and preference learning (Bai et al., 2022; Rafailov et al., 2024). Given the significant cost associated with full parameter fine-tuning, parameter-efficient fine-tuning (PEFT) (Hu et al., 2021; Chen et al., 2022; Pan et al., 2024) algorithms have emerged as a popular alternative, offering a balance between performance and resource efficiency.

Understanding what LLMs actually learn during the alignment process remains a critical question. LIMA (Zhou et al., 2023) posits that the majority of knowledge and capabilities are developed during the pretraining phase, with alignment primarily serving to refine the model's conversational style and formatting. Using a well-selected set of 1,000 training examples for supervised fine-tuning (SFT), LIMA successfully produced a high-quality aligned model. Similarly, URIAL (Lin et al., 2023) investigated the token distribution of LLMs before and after alignment and found that most changes were related to "stylistic tokens", such as discourse markers and transition words, while the knowledge-intensive content largely remained untouched, coming from the base pre-trained model. These findings imply that the alignment process mainly adjusts the model's presentation style rather than altering its foundational knowledge.

To gain a deeper understanding of LLM alignment, we adopt a distinct approach by examining it at the model parameter level. In our pilot study, we investigate the impact of different model components on alignment performance, we conducted a simple analysis by fine-tuning only specific layers and evaluating the resulting performance, as presented in Table 1. The results clearly indicate that fine-tuning different components of the LLM leads to considerable performance differences. For instance, fine-tuning the feed-forward network (FFN) layers achieves performance similar to fine-tuning all

Table 1: Impact of fine-tuning different regions of LLAMA 2-7B (Touvron et al., 2023) on alignment performance using LIMA dataset. Evaluated using MMLU (5-shot) (Hendrycks et al., 2021), GPT-4 scores on Vicuna prompts (Chiang et al., 2023), and MT-Bench prompts (Zheng et al., 2023). Fine-tuning components include query/key/value projection layers ($W_q, W_k, W_v$), output projection layer ($W_o$) in self-attention, and feed-forward networks ($W_{up}, W_{down}, W_{gate}$)

| | ATT ($W_q, W_k, W_v, W_o$) | ATT2 ($W_q, W_k, W_v$) | FFN ($W_{up}, W_{down}, W_{gate}$) | ALL (LoRA) |
|---|---|---|---|---|
| **MMLU** ↑ | 42.03 | 42.65 | 43.06 | **43.18** |
| **Vicuna** ↑ | 5.63 | 5.54 | 5.69 | **5.78** |
| **MT-Bench** ↑ | 3.82 | 3.80 | 3.92 | **3.98** |

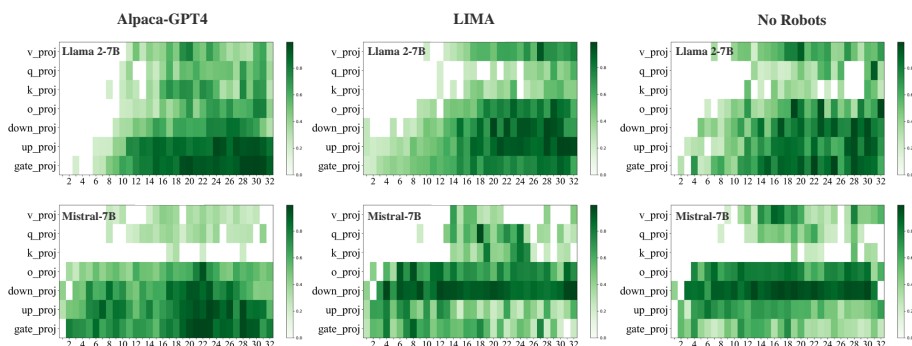

Figure 1: Layer importance ranking of LLAMA 2-7B (Touvron et al., 2023) and Mistral-7B-v0.1 (Jiang et al., 2023) by ILA across the Alpaca-GPT4 (Peng et al., 2023), LIMA Zhou et al. (2023), and No Robots (Rajani et al., 2023) datasets. Layers ranked in the top 75% by scores ($s_i$) are considered important. The x-axis represents the transformer block index, and the y-axis shows the names of linear layers within each block. The figure illustrates two key findings: (1) There is a significant overlap (up to 90%) in the important layers identified by ILA across different alignment datasets, as supported by the Jaccard similarity values in Table 2. This high consistency indicates that similar capabilities are needed for alignment, regardless of substantial differences in dataset content. (2) The important layers vary between different network architectures, suggesting that each model's structure and dynamics uniquely affect which layers are most crucial for alignment.

linear layers (i.e., with LoRA), whereas focusing solely on the attention layers causes a notable drop in performance. This observation underscores the complexity of layer-specific contributions to LLM alignment, highlighting the need for a more detailed approach to understanding their individual roles.

To this end, we propose to ***identify the layers that are most critical to alignment performance during the fine-tuning process***. We develop a novel approach for identifying the important layers for LLM alignment, called ILA. Specifically, we learn a binary mask for each incremental weight matrix in the LoRA algorithm, which serves as an indicator of layer significance. A value of zero in the binary mask indicates that the corresponding layer has negligible influence during the fine-tuning phase, while a value of one denotes that the layer is crucial for the process. We employ gradient descent to learn the binary mask effectively and offer a theoretical analysis of the optimization process. The main findings of this work are summarized as follows:

- **Consistent Layer Importance Ranking Across Different Alignment Datasets.** Despite the differences in dataset, we find similar rankings of important layers during alignment for the same pre-trained model (see Fig. 1). This suggests that the alignment process equips the model with similar capabilities, even when the training data varies significantly in both content and size. This evidence corroborates previous research findings and offers new insights into LLM alignment.

- **Enhancing Performance by Freezing Unimportant Layers.** We show that freezing approximately 25% of unimportant layers can improve model performance and that a single search for layer-wise importance ranking is sufficient for different alignment tasks within the same architecture.

- **Improving Alignment Efficiency Through Selective Fine-Tuning.** Our findings show that fine-tuning only 10-30% of the most important layers achieves performance comparable to fine-tuning

all linear layers. Additionally, integrating this approach with QLoRA allows tuning only 30-75% of key layers to maintain or enhance performance while significantly reducing resource costs.

## 2 QUANTIFYING LAYER SIGNIFICANCE IN LLM ALIGNMENT

To better understand layer significance in the alignment process of an LLM, we propose a method to identify the important layers during alignment, abbreviated as ILA. This approach involves learning a binary mask that serves as an significance indicator for each layer.

Consider a pre-trained LLM model with parameters $\boldsymbol{\theta}_0$ composed of $N$ layers, i.e., $\boldsymbol{\theta}_0 = \{\boldsymbol{\theta}_0^i\}_{i=1}^N$. The model is fine-tuned on an alignment dataset $\mathcal{D} = \{z_i\}_{i=1}^n$ with a loss function $\mathcal{L}(\boldsymbol{\theta})$. After $t$ training iterations, the model parameters are updated to $\boldsymbol{\theta}_t = \boldsymbol{\theta}_0 + \Delta\boldsymbol{\theta}_t$, where $\Delta\boldsymbol{\theta}_t$ represents the change in parameters till iteration $t$. Define a binary mask $\boldsymbol{\gamma}_t = \{\gamma_t^i | \gamma_t^i \in \{0, 1\}\}_{i=1}^N$ that encodes layer-wise importance information. We apply $\boldsymbol{\gamma}_t$ to $\Delta\boldsymbol{\theta}_t$ and define

$$\boldsymbol{\theta}_t^{\mathrm{mask}} = \boldsymbol{\theta}_0 + \boldsymbol{\gamma}_t \odot \Delta\boldsymbol{\theta}_t, \tag{1}$$

where $\odot$ is component-wise multiplication. The binary mask is applied to retain the changes in crucial layers while eliminating the rest. Below we provide a formal definition of the conditions under which training attains stability after an adequate number of iterations.

**Definition 1** ($\epsilon$-stable). *$\forall \epsilon > 0$, the model is said to be $\epsilon$-stable at iteration $T$ if, for any $t > T$, the loss function satisfies the condition*

$$|\mathbb{E}_z[\mathcal{L}(\boldsymbol{\theta}_{t+1}, z)] - \mathbb{E}_z[\mathcal{L}(\boldsymbol{\theta}_t, z)]| < \epsilon, \tag{2}$$

*where $\mathbb{E}_z[\cdot]$ denotes the expectation with respect to the alignment dataset $\mathcal{D}$.*

Once training becomes stable, we can identify the layers that are crucial for the alignment task.

**Definition 2** (Layer Importance). *The binary mask $\boldsymbol{\gamma}_t$ is defined as the solution to the following optimization problem:*

$$\boldsymbol{\gamma}_t = \underset{\boldsymbol{\gamma}_t}{\arg\min}\, \mathcal{L}(\boldsymbol{\theta}_t^{\mathrm{mask}}),\ s.t.\ \|\boldsymbol{\gamma}_t\| < H, \tag{3}$$

*where $H$ is a hyper-parameter that serves as a constraint to limit the number of important layers.*

**Efficiently Identifying the Importance Layers.** Due to the high cost of fine-tuning large models, to address the optimization problem in Eq. (3), we employ the LoRA (Hu et al., 2021) algorithm, which utilizes low-rank decomposition matrices to represent the change in model parameters till iteration $t$ ($\Delta\boldsymbol{\theta}_t$). Specifically, LoRA utilizes two trainable low-rank matrices, $\boldsymbol{B}_t^i \in \mathbb{R}^{d_i \times r_i}$ and $\boldsymbol{A}_t^i \in \mathbb{R}^{r_i \times k_i}$, to estimate the change of the $i^{\mathrm{th}}$ layer:

$$\Delta\boldsymbol{\theta}_t^i = \beta \cdot \boldsymbol{B}_t^i \boldsymbol{A}_t^i, \tag{4}$$

where $\beta$ is the scalar hyperparameter of LoRA. With the binary mask $\boldsymbol{\gamma}_t$, the $i^{\mathrm{th}}$ layer is updated by

$$\boldsymbol{\theta}_t^i = \boldsymbol{\theta}_0^i + \beta \cdot \gamma_t^i \cdot \boldsymbol{B}_t^i \boldsymbol{A}_t^i. \tag{5}$$

To ease the optimization of $\boldsymbol{\gamma}_t$, we re-parametrize each of its each components $\gamma_t^i$ as the output of a Sigmoid function, i.e., $\gamma_t^i = \sigma(s_t^i)$. Then, the update of the $i^{\mathrm{th}}$ layer becomes

$$\boldsymbol{\theta}_t^i = \boldsymbol{\theta}_0^i + \beta \cdot \sigma(s_t^i) \cdot \boldsymbol{B}_t^i \boldsymbol{A}_t^i. \tag{6}$$

Let $\boldsymbol{s}_t = \{s_t^i\}_{i=1}^N$, $\boldsymbol{\theta}_t^{\mathrm{M}} = \{\boldsymbol{\theta}_t^i\}_{i=1}^N$. The optimization problem in Eq. (3) becomes

$$\boldsymbol{s}_t = \underset{\boldsymbol{s}_t}{\arg\min}\, \mathcal{L}(\boldsymbol{\theta}_t^{\mathrm{M}}). \tag{7}$$

We use gradient descent to optimize $\boldsymbol{s}_t$. The found $s_t^i$ is considered an importance score of the $i^{\mathrm{th}}$ layer. A larger value of $s_t^i$ indicates $\gamma_t^i$ is closer to one, signifying higher importance of the $i^{\mathrm{th}}$ layer.

**Assumption 2.1** (Lipschitz-continuous). *The loss function $\mathcal{L}(\theta) : \mathbb{R}^d \to \mathbb{R}$ is continuously differentiable and L-smooth with constant $L_1 > 0$ such that*

$$\|\mathcal{L}_\infty(\theta) - \mathcal{L}(\theta')\|_2 \le L_1 \|\theta - \theta'|. \tag{8}$$

*In addition, $\mathcal{L}(\theta)$ has an L-Lipschitz continuous gradient with constant $L_2 > 0$ such that*

$$\|\nabla R(\theta) - \nabla R(\theta')\|_2 \le L_2 \|\theta - \theta'|. \tag{9}$$

---

**Algorithm 1:** Identify the Important Layers for Alignment (ILA)

---

**Input:** Pre-trained model parameters $\theta_0$, learning rate $\alpha$, the initial importance score vector
$s_0 = \{s_0^i\}_{i=1}^N$, the number of insignificant layers $K$, the low-rank matrices $A_0, B_0$ for
the LoRA algorithm.

**for** iteration $i = 1, 2, \dots$ **do**
    Update $A_t = A_{t-1} - \alpha \nabla_{A_{t-1}} \mathcal{L}(\theta_t)$ ;
    Update $B_t = B_{t-1} - \alpha \nabla_{B_{t-1}} \mathcal{L}(\theta_t)$ ;
    **if** *Training has become stable* **then**
        Solve the optimization problem in Eq. (7) by gradient descent to find $s_t = \{s_t^i\}_{i=1}^N$ ;
        Stop training;
    **end**
**end**

---

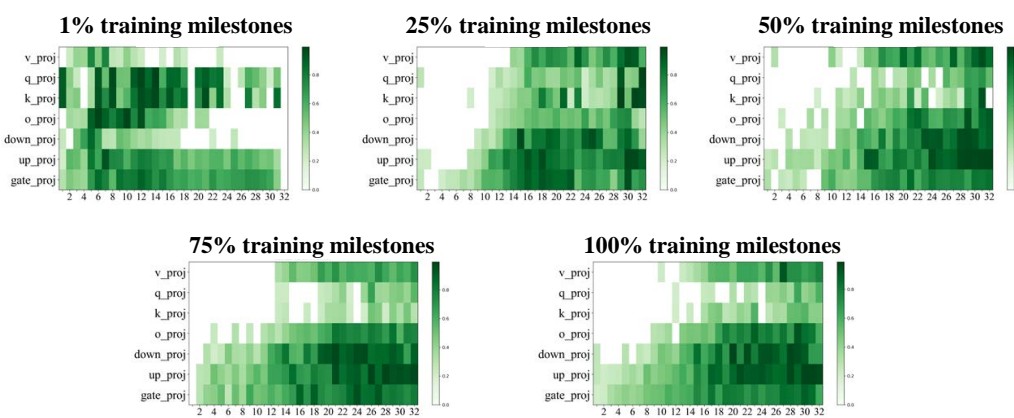

Figure 2: Layer importance ranking of LLAMA 2-7B identified by our method ILA on LIMA datasets
in different training milestones (i.e., 1%, 25%, 50%, 75%, and 100%). The x-axis represents the
transformer block index, and the y-axis shows the names of linear layers within each block. Detailed
numbers of the Jaccard similarity are presented in Table 4.

**Assumption 2.2.** *For any $t > T$, $\theta_t$ is $\epsilon$-stable. We assume there is a constant $R$ such that*

$$\|\theta_t - \theta_{t+1}\|_2 \leq R\epsilon, \tag{10}$$

*and there is a constant $Q$ such that $\|\theta_t\|_2 \leq Q$ for any $t > T$.*

**Theorem 2.1.** *For a sufficiently small $\epsilon$, $\theta_T$ is $\epsilon$-stable, thus Assumption 2.1 and Assumption 2.2 are
satisfied. For any $t > T$, we assume that $\forall i, \gamma_t^i \in [0, 1]$. Let $\gamma_t'$ denote the result of $\gamma_t$ after one step
of gradient descent, i.e., $\gamma_t' = \gamma_t - \beta \nabla_{\gamma_t} \mathcal{L}(\theta_t^{\mathrm{mask}})$. Then we have*

$$\|\gamma_t' - \gamma_{t+1}'\|_2 \leq \beta(QL_2 + L_1)R\epsilon. \tag{11}$$

This theorem demonstrates that when $\theta_T$ is $\epsilon$-stable, solving the optimization problem in Eq. (3) for
any $t > T$ yields similar results. This is because, after one step of gradient descent, the difference
between $\gamma_t$ and $\gamma_{t+1}$ is smaller than a sufficiently small number. The proof is provided in Appendix A,
and empirical results supporting this are shown in Fig. 2.

## 3 EXPERIMENTAL SETUP

**Datasets.** We train LLMs on three different alignment datasets, namely **Alpaca-GPT4** (Peng et al.,
2023), **LIMA** (Zhou et al., 2023), and **No Robots** (Rajani et al., 2023). The characteristics of each
dataset are described as follows: (1) Alpaca-GPT4 contains 52K instruction-following data generated
by GPT-4, utilizing prompts from Alpaca (Taori et al., 2023). (2) LIMA contains only 1K carefully
curated prompts and responses. (3) No Robots contains 10K instructions and demonstrations created
by skilled human annotators.

Table 2: Jaccard similarities of important layers identified during fine-tuning of LLAMA 2-7B and Mistral-7B on various datasets. Top 75% highest-scoring layers are determined as important layers.

| Datasets | LLAMA 2-7B | | | Mistral-7B | | |
|---|---|---|---|---|---|---|
| | LIMA | No Robots | Alpaca-GPT4 | LIMA | No Robots | Alpaca-GPT4 |
| LIMA | - | - | - | - | - | - |
| No Robots | 0.91 | - | - | 0.90 | - | - |
| Alpaca-GPT4 | 0.90 | 0.90 | - | 0.89 | 0.93 | - |

Figure 3: Jaccard similarities of important layers identified during fine-tuning of LLAMA 2-7B on the LIMA dataset with varying random seeds. The top 75% highest-scoring layers are designated as important layers.

| Random Seed | seed1 | seed2 | seed3 |
|---|---|---|---|
| seed1 | - | - | - |
| seed2 | 0.92 | - | - |
| seed3 | 0.91 | 0.91 | - |

Figure 4: Jaccard similarities between sets of important layers identified at different milestones during the fine-tuning of LLAMA 2-7B on the LIMA dataset. The top 75% highest-scoring layers are designated as important layers for this analysis.

| Training Milestones | 1% | 25% | 50% | 75% | 100% |
|---|---|---|---|---|---|
| 1% | - | - | - | - | - |
| 25% | 0.69 | - | - | - | - |
| 50% | 0.70 | 0.91 | - | - | - |
| 75% | 0.69 | 0.90 | 0.92 | - | - |
| 100% | 0.69 | 0.91 | 0.92 | 0.93 | - |

**Models and Baselines.** We use four different models as the base for our experiments: LLAMA 2-7B (Touvron et al., 2023), LLAMA 2-13B (Touvron et al., 2023), Llama 3.1-8B (Dubey et al., 2024), and Mistral-7B-v0.1 (Jiang et al., 2023). The baselines include (1) **LoRA** (Hu et al., 2021): We add trainable pairs of rank decomposition matrices in parallel to existing weight matrices, including query/key/value projection ($W_q$, $W_k$, $W_v$), output projection ($W_o$) in the self-attention, feed-forward networks ($W_{\text{up}}$, $W_{\text{down}}$, $W_{\text{gate}}$), and the output layer on top of the transformer ($W_{\text{head}}$). (2) **AdaLoRA** (Zhang et al., 2023a): It dynamically adjusts the rank of incremental matrices to control the parameter budget. Similar to LoRA, we add AdaLoRA modules to all linear layers of the base model. (3) **QLoRA** (Dettmers et al., 2023): It is a fine-tuning method that significantly reduces memory usage by quantizing the weights of pre-trained language models while maintaining competitive performance. (4) **Full Finetune**: The model is initialized to the pre-trained weights and biases, and all model parameters undergo gradient updates.

**Evaluation and Training Setup.** Our evaluation of language model alignment encompasses two main dimensions: (1) **Language Understanding Ability**: We utilized three distinct datasets (i.e., **MMLU** (Massively Multitask Language Understanding) (Hendrycks et al., 2021) and **Hellaswag** (Zellers et al., 2019) to evaluate this aspect. MMLU evaluates models across diverse subjects requiring specialized knowledge, while Hellaswag tests commonsense reasoning by asking the model to predict the most plausible continuation of a given context. (2) **Conversational Ability**: We use two different datasets: **MT-Bench** (Zheng et al., 2023), which involves multi-turn conversations, and **Vicuna** (Chiang et al., 2023), which involves single-turn conversations. We use GPT-4 to score the responses. We asks GPT-4 to grade and give a score to model's answer directly without pairwise comparison, using the implementation version of MT-Bench (Zheng et al., 2023). For a fair comparison, we conduct a small range of training hyperparameter searches for LoRA and full fine-tuning to ensure that we get strong baselines. More details are provided in Appendix B.

**Targeted Performance.** (1) **Language Understanding Ability**: Recent research (Du et al., 2020; Sun et al., 2021; Dubey et al., 2024) suggests that the learning of language understanding tasks essentially occurs during the pre-training phase of the base model. Therefore, significant performance improvements in language understanding tasks (i.e., MMLU, Hellaswag) after alignment are not expected. However, *it is crucial to ensure the model retains the learned knowledge during alignment*. (2) **Conversational Ability**: Without alignment, the pre-train model's conversational ability is poor. For example, LLAMA 2-7B often produces incorrect or irrelevant responses on the Vicuna dataset. However, *its conversational ability can be significantly improved through the alignment process*.

Table 3: Comparative evaluation of LLAMA 2-7B and Mistral-7B-v0.1 models finetuned on the No Robots Dataset. This table presents the 5-shot test accuracy for the MMLU benchmark, alongside the 0-shot test accuracy for the Hellaswag dataset. Cells highlighted in grey indicate that ILA has enhanced the performance of the base model. The best result is marked in bold.

| Models | Methods | Language Understanding | | Conversational Ability | |
|---|---|---|---|---|---|
| | | MMLU ↑ | Hellaswag ↑ | Vicuna ↑ | MT-Bench ↑ |
| LLAMA 2-7B | AdaLoRA | 45.23 | 57.30 | 5.70 | 4.05 |
| | Full Finetune | 45.72 | 57.69 | 6.00 | 3.93 |
| | Full Finetune w/ ILA | **45.98** | 57.87 | 5.90 | 4.21 |
| | LoRA | 44.58 | 59.46 | 6.23 | 4.70 |
| | LoRA w/ ILA | 45.78 | **59.65** | **6.30** | **4.93** |
| Mistral-7B-v0.1 | AdaLoRA | 62.13 | 61.68 | 6.10 | 5.03 |
| | Full Finetune | 61.05 | **64.26** | 6.70 | 5.56 |
| | Full Finetune w/ IFILA | 61.75 | 64.21 | 6.73 | **5.70** |
| | LoRA | 61.95 | 62.90 | 6.77 | 5.35 |
| | LoRA w/ IFILA | **62.14** | 62.80 | **6.82** | 5.42 |

## 4 EMPIRICAL FINDINGS

### 4.1 LAYER SIGNIFICANCE IN LLM ALIGNMENT

In this subsection, we applied our ILA algorithm to identify the ranking of important layers during alignment across three different datasets: No Robots, LIMA, and Alpaca-GPT4, as shown in Fig. 1. Additionally, we analyzed the importance ranking of layers identified at different training milestones, as depicted in Fig. 2. To further validate the similarity of these important layers, we used the Jaccard similarity coefficient to quantify the relationship between two sets. Specifically, we defined the top 75% highest-scoring layers as the important layers, denoted as set $S$. The similarity between two distinct sets, $S_1$ and $S_2$, is calculated as: $J(S_1, S_2) = \frac{|S_1 \cap S_2|}{|S_1 \cup S_2|}$. A value of $J = 1$ indicates identical sets, while $J = 0$ indicates no overlap. Below, we highlight our main observations.

> **Consistency in Layer Importance Ranking Across Various Alignment Datasets.** Our findings demonstrate a remarkable consistency in layer importance ranking, as evidenced by: (1) the retrieval of highly similar important layers across different alignment datasets, as shown in Fig. 1 and Table 2; (2) the consistent identification of important layers despite the optimization of $\gamma$ with varying random seeds, as illustrated in Table 3; (3) the ability to identify similar important layers at different or early (25%) training stages, as depicted in Fig. 2 and Table 4.

The experimental results corroborate the robustness of our algorithm, which consistently identifies stable and similar layers across different alignment datasets. This is particularly noteworthy in light of recent work that suggests alignment fundamentally involves shifts in stylistic tokens (Lin et al., 2023). *Thus, the essence of alignment is the pursuit of similar capabilities, which aligns with our discovery that the important layers corresponding to different datasets exhibit similarity.* This convergence of findings underscores the intrinsic alignment of our algorithm's performance with the fundamental objectives of dataset alignment.

Given the established importance ranking of the model layers, which proves stable for the alignment task, we must consider how to leverage this ranking. We will address this from both performance and efficiency perspectives. First, to maximize the performance of the fine-tuned model, we should avoid fine-tuning layers that could negatively impact the model, focusing instead on those deemed less significant. Second, to enhance the efficiency of fine-tuning and minimize resource consumption, we should concentrate our efforts on layers that are particularly vital to the model's success. Detailed experiments and analyses of these two cases will be presented in the following section.

### 4.2 ENHANCING ALIGNMENT PERFORMANCE THROUGH FREEZING UNIMPORTANT LAYERS

To achieve optimal model performance, we excluded the unimportant layers, specifically those whose modifications would negatively impact fine-tuning. Approximately 25% of the unimportant layers

Table 4: Comparative evaluation of LLAMA 2-7B and Mistral-7B-v0.1 models finetuned on the LIMA Dataset. This table presents the 5-shot test accuracy for the MMLU benchmark, alongside the 0-shot test accuracy for the Hellaswag dataset. Cells highlighted in grey indicate that ILA has enhanced the performance of the base model. The best result is marked in bold.

| Models | Methods | Language Understanding | | Conversational Ability | |
|---|---|---|---|---|---|
| | | MMLU ↑ | Hellaswag ↑ | Vicuna ↑ | MT-Bench ↑ |
| LLAMA 2-7B | AdaLoRA | 44.21 | 59.85 | 5.66 | 3.82 |
| | Full Finetune | **46.36** | 62.06 | 5.85 | 3.91 |
| | Full Finetune w/ ILA | 46.32 | **62.18** | **5.96** | 4.02 |
| | LoRA | 43.18 | 54.52 | 5.78 | 3.98 |
| | LoRA w/ ILA | 44.13 | 54.55 | 5.88 | **4.10** |
| Mistral-7B-v0.1 | AdaLoRA | **62.40** | 61.52 | 6.58 | 4.46 |
| | Full Finetune | 60.11 | 63.76 | **6.99** | 5.39 |
| | Full Finetune w/ ILA | 61.01 | 64.01 | 6.94 | **5.47** |
| | LoRA | 60.83 | 65.42 | 6.82 | 4.88 |
| | LoRA w/ ILA | 61.52 | **65.51** | 6.92 | 5.34 |

Table 5: Results of fine-tuning Mistral-7B-v0.1 on the No Robots dataset. This table presents the 5-shot test accuracy for the MMLU benchmark, along with the 0-shot test accuracy for the Hellaswag dataset. The percentages in parentheses indicate the proportion of important linear layers fine-tuned relative to all linear layers. The best results are highlighted in bold.

| Models | Methods | Language Understanding | | Conversational Ability | |
|---|---|---|---|---|---|
| | | MMLU ↑ | Hellaswag ↑ | Vicuna ↑ | MT-Bench ↑ |
| Mistral-7B-v0.1 | LoRA | **61.95** | **62.90** | **6.77** | 5.35 |
| | LoRA w/ ILA (10%) | 62.09 | 61.94 | 6.49 | 5.08 |
| | LoRA w/ ILA (20%) | 61.83 | 62.16 | 6.60 | 5.23 |
| | LoRA w/ ILA (30%) | 61.89 | 62.79 | 6.71 | **5.37** |

were removed. The main results on **No Robots** and **LIMA** are presented in Table 3 and Table 4 respectively. For additional results of **LLAMA 2-13B** and main results on **Alpaca-GPT4** dataset, please refer to Appendix C. Based on the results, we highlight two key observations:

> (1) **Freezing Unimportant Layers May Enhance Performance.** Compared to LoRA and full fine-tuning, ILA consistently outperformed in most evaluation metrics while matching performance in others. Freezing approximately 25% of unimportant layers yielded better results than tuning all layers. (2) **Only a Single Search for Layer-wise Importance Ranking is Required for a Given Network Architecture.** The importance ranking was remarkably stable across alignment tasks for a given architecture, allowing us to compute the ranking on the No Robots dataset and apply it effectively to other datasets.

The results indicate that ILA provides robust and efficient fine-tuning by focusing only on significant layers while excluding those that negatively impact the model. When compared to AdaLoRA, even though we explored a narrow range of the hyperparameter $t_r$ (target average rank of incremental matrices), our method performed better. This outcome highlights that simply adjusting LoRA's matrix rank does not necessarily yield superior results in alignment tasks, as confirmed by other studies (Dettmers et al., 2023).

Furthermore, as discussed in Section 4.1, the stability of the layer importance ranking across various alignment datasets suggests that it is often sufficient to conduct a single importance ranking search for a given network architecture. In our experiments, we computed the layer importance ranking using full training iterations on the No Robots dataset, and then directly applied this ranking to other datasets. Although dataset-specific importance rankings can yield further improvements (see Table. 9 in Section 5), the consistent cross-dataset performance achieved using a single ranking highlights the robustness and generalizability of our approach.

Table 6: Comparison of fine-tuning results using QLoRA on LLAMA 2-7B and Llama 3.1-8B versus QLoRA applied to selected important layers identified by ILA. This table shows the 5-shot test accuracy for the MMLU benchmark and the 0-shot test accuracy for the Hellaswag dataset. Cells highlighted in grey indicate performance improvements achieved by ILA over the base model.

| Datasets | Methods | Language Understanding | | Conversational Ability | |
|---|---|---|---|---|---|
| | | MMLU ↑ | Hellaswag ↑ | Vicuna ↑ | MT-Bench ↑ |
| LIMA | LoRA | 53.85 | 63.08 | 6.40 | 4.43 |
| | LoRA w/ ILA (75%) | 54.33 | 62.04 | 6.54 | 4.55 |
| | LoRA w/ ILA (30%) | 54.27 | 62.88 | 6.31 | 4.54 |
| NoRobots | LoRA | 54.08 | 61.73 | 6.69 | 4.94 |
| | LoRA w/ ILA | 54.45 | 61.13 | 6.77 | 5.05 |

Table 7: GPU memory usage for LoRA, QLoRA, and LoRA/QLoRA with only 30% of important layers fine-tuned. Batch size is set to 2, and the maximum token length is 1024. Percentages in parentheses indicate the proportion of linear layers fine-tuned.

| | LoRA (100%) | LoRA w/ ILA (30%) | QLoRA (100%) | QLoRA w/ ILA (30%) |
|---|---|---|---|---|
| GPU Memory Usage (MiB) | 32988 | **25614** | 26032 | **18142** |

### 4.3 ENHANCING ALIGNMENT EFFICIENCY BY ONLY FINE-TUNING THE CRITICAL LAYERS

To investigate this issue, we fine-tuned only 10%, 20%, and 30% of the important layers of Mistral-7B-v0.1, as identified by ILA, on the No Robots dataset, and compared the results with the LoRA algorithm. The results demonstrate clear benefits in focusing on a subset of important layers:

> (1) **Fine-Tuning a Small Subset of Important Layers Achieves Competitive Performance and Enhances Efficiency.** Fine-tuning the top 10% or 20% of important layers results in only a slight performance drop compared to full fine-tuning, while fine-tuning 30% of the parameters nearly matches the performance of full fine-tuning (see Table 5). This demonstrates that focusing on a small, carefully selected subset of important layers is sufficient for efficient fine-tuning without significant performance loss. (2) **Our Method Can be Applied to Enhance QLoRA, Further Reducing Cost.** By integrating our method with QLoRA, we fine-tuned only about 30-75% of the key layers while maintaining or improving model performance (see Table 6). This highlights the efficiency of our approach, achieving comparable or superior results with significantly fewer layers involved.

These findings underline the robustness of our layer selection strategy, allowing efficient use of resources with minimal trade-offs in performance. Additionally, our integration with QLoRA confirms that fine-tuning only a targeted subset of important layers enhances both the performance and efficiency of state-of-the-art methods in reducing memory usage during fine-tuning.

To provide a more intuitive understanding of how our method reduces GPU memory usage, we measured the memory consumption of QLoRA, LoRA, and the versions that fine-tune only a subset of important layers identified by ILA in Table 7. The results show that our method reduces GPU memory requirements while maintaining competitive performance, making it an effective strategy for resource-constrained environments.

### 4.4 ABLATION STUDY

**Observation 1: Randomly or manually selecting layers for fine-tuning does not work.**

To substantiate the accuracy and efficacy of the ranking and importance layers identified by our algorithm, we contrast the baseline that optimizes all linear layers without any freezing with three alternative scenarios: (1) **RL** 1 and **RL** 2, where the top-$K$ layers to be frozen are randomly selected using two different random seeds; (2) **FL**, which involves freezing the first $K$ linear layers; and (3) **LL**, which entails freezing the last $K$ linear layers. The experimental results indicate that neither

Table 8: Performance comparison of ILA, random layer selection, and position-based layer selection for fine-tuning LLAMA 2-7B on the No Robots Dataset. The abbreviations **RL** 1 and **RL** 2 refer to the approach of **r**andomly selecting $K$ **l**ayers to freeze during the fine-tuning process, with each employing a distinct random seed. **FL** denotes the strategy of freezing the **f**irst $K$ **l**ayers, while **LL** indicates the freezing of the **l**ast $K$ **l**ayers. Performance reductions compared with our ILA algorithm are highlighted in blue.

| Methods | Language Understanding | | Conversational Ability | |
|---|---|---|---|---|
| | MMLU ↑ | Hellaswag ↑ | Vicuna ↑ | MT-Bench ↑ |
| LoRA | 44.58 | 59.46 | 6.23 | 4.70 |
| LoRA w/ RL 1 | 44.23 | 59.71 | 6.08 | 4.60 |
| LoRA w/ RL 2 | 43.98 | 59.11 | 6.10 | 4.68 |
| LoRA w/ FL | 44.02 | 59.32 | 6.13 | 4.59 |
| LoRA w/ LL | 44.61 | 59.21 | 6.20 | 4.63 |
| LoRA w/ ILA | 45.78 | 59.65 | 6.30 | 4.93 |

Table 9: Results of fine-tuning Mistral-7B-v0.1 on the LIMA dataset using ILA to identify important layers from various datasets. **Dataset (Imp. Layers)** indicates the datasets utilized to search for the important layers. **Intersection** represents freezing the layers that are the intersection of the top-$K$ least important layers found from the LIMA, No Robots, and Alpaca GPT4 datasets.

| Dataset (Imp. Layers) | Dataset (Finetune) | Language Understanding | | Conversational Ability | |
|---|---|---|---|---|---|
| | | MMLU ↑ | Hellaswag ↑ | Vicuna ↑ | MT-Bench ↑ |
| LIMA | LIMA | **61.82** | 65.48 | 6.99 | 5.38 |
| No Robots | LIMA | 61.52 | 65.51 | 6.92 | 5.34 |
| Alpaca-GPT4 | LIMA | 61.23 | 65.20 | 7.03 | 5.21 |
| Intersection | LIMA | 61.49 | **65.62** | **7.06** | **5.44** |

the random freezing of $K$ layers nor the selective freezing of either the first or last $K$ linear layers could outperform the baseline of tuning all layers on most evaluation metrics. In contrast, our ILA can accurately identifies the layers of importance and freeze the top-$K$ least important layers, thereby achieving substantial improvements. This demonstrates that ILA effectively pinpoints the non-critical layers for freezing, optimizing the fine-tuning process and enhancing model performance without the need to adjust every layer.

**Observation 2: Cross-dataset evaluation of layer importance can lead to the best results.**

As indicated in Table 2, subtle differences are observed in the important layers identified across various datasets. This observation leads to an intuitive hypothesis that layers consistently deemed unimportant across all datasets may truly be non-essential. To this end, we intersect the top-$K$ least important layers from three distinct datasets (i.e., LIMA, No Robots, and Alpaca-GPT4) to determine the ultimately non-critical layers. These layers are subsequently frozen during fine-tuning, with the specific outcomes presented in Table 9.

Our analysis reveal that a holistic consideration of layer importance across multiple datasets yields superior results compared to dataset-specific approaches. For instance, identifying important layers within the LIMA dataset and fine-tuning on the No Robots dataset is less effective than an integrated approach. Similarly, finding important layers and fine-tuning exclusively on the No Robots dataset do not perform as well as the comprehensive method. This suggests that a cross-dataset evaluation of layer importance can lead to more robust and effective fine-tuning strategies.

**Observation 3: The computation cost of ILA is low.**

Our ILA algorithm consists of two stages. **Stage 1**: We use LoRA to train the model until it is sufficiently stable, i.e., $\epsilon$-stable. **Stage 2**: We fix the backbone network and the LoRA modules to learn the importance weights ($\gamma_t$). For LLAMA 2-7B and Mistral-7B-v0.1, $|\gamma_t| = 225$. To quantify computation cost, we measured the training time per iteration for LLAMA 2-7B in stages 1 and 2

with a batch size of 32. For stage 1, the training time is **6671 ms**. For stage 2, the training time is **5343 ms**. In Stage 2, we train for **128 batches** on each dataset. Therefore, we only tune the model for about $5.34 \times 128 \div 60 \approx 11$ miniutes. The main training cost is in Stage 1. However, as shown in Table 4, it is not necessary to complete the entire training process; reaching $25\% \sim 50\%$ of the training milestones is sufficient.

## 5 RELATED WORKS

**Large Language Models (LLMs) Alignment.** Language models are initially pretrained to learn general-purpose representations, enabling their transfer to a wide range of language understanding and generation tasks (Qiu et al., 2024; Jiang et al., 2024; Nijkamp et al., 2022). To align these models with specific user needs and improve their performance on targeted applications, techniques such as *Instruction Tuning* (Zhang et al., 2023c; Sun et al., 2023; Muennighoff et al., 2023) and *Preference Learning* (Hejna et al., 2023; Guan et al., 2022; Rafailov et al., 2024; Song et al., 2024; Li et al., 2024) are commonly employed. Tuning-based alignment can introduce issues such as forgetting in LLMs (Wang et al., 2022a;b) and underfitting (Zhang et al., 2023c; Sun et al., 2023).

To explore the nature of model alignment through various studies. LIMA (Zhou et al., 2023) achieved a well-aligned model by fine-tuning nearly 1,000 samples using SFT, and hypothesized that the alignment process essentially teaches the model how to conduct conversations in specific formats or meet certain requirements without acquiring new knowledge. Similar findings have been reported in recent studies (Chen et al., 2023; Lee et al., 2023; Gudibande et al., 2023). Duan et al. (2023) analyzed the hidden states of LLMs, exploring the similarities between in-context learning (ICL) and instruction tuning (IT) regarding their impact on downstream tasks. URIAL (Lin et al., 2023) investigated the token distribution before and after alignment, suggesting that alignment primarily shifts "stylistic tokens" like discourse markers and transition words, while the distribution of knowledge-intensive terms remains largely unchanged. Based on prior research, we hypothesize that the abilities learned during alignment are relatively narrow in scope. To better understand this process, we propose an approach to identify which layers are genuinely important during alignment.

**Parameter Efficient Fine-Tuning (PEFT).** To tackle the high computational costs of full-model fine-tuning, especially with Pre-trained Language Models (PLMs) ranging from billions to trillions of parameters (Brown et al., 2020; Fedus et al., 2022), PEFT methods have been developed to reduce parameter usage while maintaining the effectiveness and stability of knowledge transfer (Tang et al., 2024; Peng et al., 2024). These approaches include partial fine-tuning, which selectively targets specific model components (Zaken et al., 2021; Zhao et al., 2020; Ansell et al., 2021; Guo et al., 2020), and soft prompt-based fine-tuning (Lester et al., 2021; Li & Liang, 2021; Asai et al., 2022). Notable methods include BitFit (Zaken et al., 2021), Adapter (Houlsby et al., 2019), LoRA (Hu et al., 2021) and its variants (Zhang et al., 2023b; Meng et al., 2024). Recent studies (Pan et al., 2024; Xu & Zhang, 2024; Panda et al., 2024) have shown that fine-tuning only a small portion of a model while masking most components can still achieve promising results in LLMs. However, these masking strategies are often applied randomly, akin to dropout, which is suboptimal and lacks consistency. While effective for efficient fine-tuning, these methods provide limited insight into understanding the alignment task. To overcome these limitations, our approach leverages the concept of skill localization (Panigrahi et al., 2023) by dynamically identifying and fine-tuning the critical components for each task. By focusing solely on the most important regions, this method significantly improves the efficiency of model fine-tuning while ensuring strong performance.

## 6 CONCLUSIONS

In conclusion, our proposed method, ILA, focuses on identifying critical layers in the alignment process by learning binary masks for LoRA weight matrices. ILA demonstrates consistent identification of important layers across different datasets, regardless of significant content variations, suggesting that the alignment process imparts similar capabilities to the model irrespective of the training data. This finding provides valuable insights into the specific roles of layers during alignment. By strategically tuning only the most vital layers, ILA effectively reduces computational overhead, and by freezing less important layers, it further enhances model responsiveness and accuracy, leading to more efficient resource utilization.

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

## A    PROOF OF THEOREM 2.1

**Theorem A.1.** *For a sufficiently small $\epsilon$, $\theta_T$ is $\epsilon$-stable, thus Assumption 2.1 and Assumption 2.2 are satisfied. For any $t > T$, we assume that $\forall i, \gamma_t^i \in [0, 1]$. Let $\gamma_t'$ denote the result of $\gamma_t$ after one step of gradient descent, i.e., $\gamma_t' = \gamma_t - \beta \nabla_{\gamma_t} \mathcal{L}(\theta_t^{\text{mask}})$. Then we have*

$$\|\gamma_t' - \gamma_{t+1}'\|_2 \leq \beta(QL_2 + L_1)R\epsilon. \tag{12}$$

*Proof.* Let $\hat{\gamma}$ be the initial values of $\gamma_t$ and $\gamma_{t+1}$. Then we have

$$\gamma_t' = \hat{\gamma} - \beta \nabla_{\gamma_t} \mathcal{L}(\theta_t^{\text{mask}}) \tag{13}$$

$$\gamma_{t+1}' = \hat{\gamma} - \beta \nabla_{\gamma_{t+1}} \mathcal{L}(\theta_{t+1}^{\text{mask}}) \tag{14}$$

The difference of $\gamma_t'$ and $\gamma_{t+1}'$ is

$$\|\gamma_t' - \gamma_{t+1}'\|_2 = \|(\hat{\gamma} - \beta \nabla_{\gamma_t} \mathcal{L}(\theta_t^{\text{mask}})) - (\hat{\gamma} - \beta \nabla_{\gamma_{t+1}} \mathcal{L}(\theta_{t+1}^{\text{mask}}))\|_2 \tag{15}$$

$$= \beta \|\nabla_{\gamma_t} \mathcal{L}(\theta_t^{\text{mask}}) - \nabla_{\gamma_{t+1}} \mathcal{L}(\theta_{t+1}^{\text{mask}})\|_2 \tag{16}$$

$$= \beta \|\theta_t \odot \nabla_{\theta_t^{\text{mask}}}(\theta_t^{\text{mask}}) - \theta_{t+1} \odot \nabla_{\theta_{t+1}^{\text{mask}}}(\theta_{t+1}^{\text{mask}})\|_2 \tag{17}$$

$$\leq \beta \|\theta_t \odot \left( \nabla_{\theta_t^{\text{mask}}}(\theta_t^{\text{mask}}) - \nabla_{\theta_{t+1}^{\text{mask}}}(\theta_{t+1}^{\text{mask}}) \right) \|_2 \tag{18}$$

$$+ \beta \|(\theta_t - \theta_{t+1}) \odot \nabla_{\theta_{t+1}^{\text{mask}}}(\theta_{t+1}^{\text{mask}})\|_2. \tag{19}$$

Because $\mathcal{L}(\theta)$ has an L-Lipschitz continuous gradient with constant $L_2 > 0$, and $\|\theta_t\| \leq Q$,

$$\|\theta_t \odot \nabla_{\theta_t^{\text{mask}}}(\theta_t^{\text{mask}}) - \theta_{t+1} \odot \nabla_{\theta_{t+1}^{\text{mask}}}(\theta_{t+1}^{\text{mask}})\|_2 \leq QL_2 \|\theta_t^{\text{mask}} - \theta_{t+1}^{\text{mask}}\|_2 \tag{20}$$

$$= QL_2 \|\Delta\theta_{t+1} - \Delta\theta_t\|_2 \tag{21}$$

$$= QL_2 \|\theta_{t+1} - \theta_t\|_2 \tag{22}$$

Because $\mathcal{L}(\theta)$ is L-smooth with constant $L_1$,

$$\|(\theta_t - \theta_{t+1}) \odot \nabla_{\theta_{t+1}^{\text{mask}}}(\theta_{t+1}^{\text{mask}})\|_2 \leq L_1 \|\theta_t - \theta_{t+1}\|. \tag{23}$$

Therefore,

$$\|\gamma_t' - \gamma_{t+1}'\|_2 \leq \beta(QL_2 + L_1)\|\theta_t - \theta_{t+1}\|_2. \tag{24}$$

According to the Assumption 2.2, we have $\|\theta_t - \theta_{t+1}\|_2 \leq R\epsilon$, hence,

$$\|\gamma_t' - \gamma_{t+1}'\|_2 \leq \beta(QL_2 + L_1)R\epsilon. \tag{25}$$

$\square$

## B    EXPERIMENTAL SETUP

For all experiments, we follow fine-tuning hyperparameters: we use AdamW with $\beta_1 = 0.9$, $\beta_2 = 0.99$ and weight decay of $0.1$. The scheduler employed is a cosine scheduler with a warmup ratio of $0.01$. For LoRA baselines, we set the hyperparameter rank $r$ as 32.

### B.1    NO ROBOTS DATASET

We do a hyperparameter search for LoRA over the following variables: learning rate $\{0.001, 0.002, 0.0005, 0.0002, 0.0001\}$, training epochs $\{2, 3, 4, 5\}$. We do hyperparameter search for full fine-tuning over the following variables: learning rate $\{1e-4, 2e-5, 1e-5, 5e-6, 2e-6\}$, training epochs $\{2, 3, 4, 5\}$.

**LLAMA 2-7B**. Both LoRA and AdaLoRA use a dropout rate of $0.1$ and a learning rate of $0.001$. The number of training epochs is 3. For full fine-tuning, the learning rate is set to $0.00001$, with the number of training epochs also being 3. The training parameters for IFILA are consistent with those of the baselines.

**Mistral-7B.** For LoRA and AdaLorA, we set the dropout rate as $0.1$. The learning is $0.0002$. The number of training epochs is 2. For full fine-tuning, the learning rate is set as $0.000002$ and the number of training epochs is 2. The training parameters of IFILA are the same as the baselines.

Table 10: Fine-tuning results of LLAMA 2-13B on the LIMA and No Robots datasets. This table shows the 5-shot test accuracy for the MMLU benchmark along with the 0-shot test accuracy for the Hellaswag dataset. Cells highlighted in grey indicate that ILA has improved the performance of the base model.

| Datasets | Methods | Language Understanding | | Conversational Ability | |
|---|---|---|---|---|---|
| | | MMLU ↑ | Hellaswag ↑ | Vicuna ↑ | MT-Bench ↑ |
| LIMA | LoRA | 53.85 | 63.08 | 6.40 | 4.43 |
| | LoRA w/ ILA | 54.33 | 62.04 | 6.54 | 4.55 |
| No Robots | LoRA | 54.08 | 61.73 | 6.69 | 4.94 |
| | LoRA w/ ILA | 54.45 | 61.13 | 6.77 | 5.05 |

## B.2 LIMA DATASET

We do a hyperparameter search for LoRA over the following variables: learning rate $\{0.001, 0.002, 0.0005, 0.0002, 0.0001\}$, training epochs $\{5, 10, 15, 20\}$. We do hyperparameter search for full fine-tuning over the following variables: learning rate $\{1e - 4, 2e - 5, 1e - 5, 5e - 6, 2e - 6\}$, training epochs $\{5, 10, 15, 20\}$.

**LLAMA 2-7B** . For LoRA and AdaLorA, we set the dropout rate as $0.1$. The learning is $0.001$. The number of training epochs is $20$. For full fine-tuning, the learning rate is set as $0.00001$ and the number of training epochs is $5$. The training parameters of IFILA are the same as the baselines.

**Mistral-7B.** For LoRA and AdaLorA, we set the dropout rate as $0.1$. The learning is $0.0002$. The number of training epochs is $5$. For full fine-tuning, the learning rate is set as $0.000005$ and the number of training epochs is $5$. The training parameters of IFILA are the same as the baselines.

## B.3 ALPACA-GPT DATASET.

We do a hyperparameter search for LoRA over the following variables: learning rate $\{0.001, 0.002, 0.0005, 0.0002, 0.0001\}$, training epochs $\{0.5, 1, 1.5, 2, 3\}$. We do hyperparameter search for full fine-tuning over the following variables: learning rate $\{1e - 4, 2e - 5, 1e - 5, 5e - 6, 2e - 6\}$, training epochs $\{0.5, 1, 1.5, 2, 3\}$.

**LLAMA 2-7B** . For LoRA and AdaLorA, we set the dropout rate as $0.1$. The learning is $0.0002$. The number of training epochs is $1.5$. For full fine-tuning, the learning rate is set as $0.000002$ and the number of training epochs is $0.5$. The training parameters of IFILA are the same as the baselines.

**Mistral-7B.** For LoRA and AdaLorA, we set the dropout rate as $0.1$. The learning is $0.0002$. The number of training epochs is $5$. For full fine-tuning, the learning rate is set as $0.000002$ and the number of training epochs is $0.5$. The training parameters of IFILA are the same as the baselines.

## C ADDITIONAL EXPERIMENTS

### C.1 ADDITIONAL EXPERIMENTS ON MODEL SCALABILITY

To assess whether freezing unimportant layers continues to enhance model performance at a larger scale, we conducted additional experiments on LLAMA 2-13B. Specifically, we fine-tuned LLAMA 2-13B using the No Robots and LIMA datasets, with results compared against LoRA presented in the table below. The experimental outcomes demonstrate that our method maintains strong performance on LLAMA 2-13B. Despite the increased model size, the underlying architectural similarities suggest that our approach remains effective and scalable, likely extending its benefits to even larger models.

Table 11: Comparative Evaluation of LLAMA 2-7B and Mistral-7B-v0.1 Models finetuned on the Alpaca-GPT4 Dataset. This table presents the 5-shot test accuracy for the MMLU benchmark, alongside the 0-shot test accuracy for the Hellaswag dataset. Cells highlighted in grey indicate that ILA has enhanced the performance of the base model. The best result is marked in bold.

| Models | Methods | Language Understanding | | Conversational Ability | |
|---|---|---|---|---|---|
| | | MMLU ↑ | Hellaswag ↑ | Vicuna ↑ | MT-Bench ↑ |
| LLAMA 2-7B | AdaLoRA | 46.13 | 57.85 | **7.06** | 3.90 |
| | Full Finetune | 45.91 | 57.73 | 4.62 | 3.56 |
| | Full Finetune w/ ILA | **46.23** | 57.67 | 5.03 | 4.01 |
| | LoRA | 43.66 | **58.49** | 6.91 | 4.21 |
| | LoRA w/ ILA | 44.69 | 58.22 | 7.01 | **4.58** |
| Mistral-7B-v0.1 | AdaLoRA | **62.48** | 62.08 | 7.43 | 5.51 |
| | Full Finetune | 60.56 | 62.80 | 4.55 | 3.82 |
| | Full Finetune w/ ILA | 60.88 | **62.91** | 5.22 | 4.11 |
| | LoRA | 61.82 | 62.70 | 7.31 | 6.15 |
| | LoRA w/ ILA | 62.14 | 62.80 | **7.45** | **6.19** |

We also carried out further experiments on **Alpaca-GPT4** using **LLAMA 2-7B** and **Mistral-7B-v0.1** to evaluate the adaptability of our approach across different model architectures. Consistently, our method outperformed LoRA while requiring fewer layers to be fine-tuned. These findings further validate the robustness and scalability of our approach, showing its capability to effectively enhance performance across various model sizes and architectural variations.

