# OpenReview forum: "Understanding Layer Significance in LLM Alignment"
_ICLR.cc/2025/Conference — ICLR 2025 Conference Withdrawn Submission_

### Official Review · Reviewer_epzj · 2024-10-30

**Soundness:** 3
**Presentation:** 2
**Contribution:** 2
**Rating:** 5
**Confidence:** 3

**Summary:**

This paper proposes an ILA (identify the important layers for LLM alignment) method to identify the most relevant layers for alignment tasks, then improves performance by freezing irrelevant layers. The authors focus on Alignment tasks, including LIMA, Alpaca, and no robots.

**Strengths:**

On one hand, the authors provide detailed motivation in the introduction explaining why freezing different layers during training is necessary, which can help reduce catastrophic forgetting to some extent. On the other hand, the paper offers comprehensive empirical evidence showing their proposed method achieves generally improved performance across LoRA, AdaLoRA, QLoRA and full fine-tuning.

**Weaknesses:**

Indeed, this paper's novelty is limited. The core motivation and main method of freezing certain layers to avoid overfitting was proposed three years ago in paper [1], which even provided finer-grained control over the degree of parameter freezing. In my view, the authors merely validated this approach on Alignment tasks (just one type of fine-tuning task). While I acknowledge the technical implementations differ, given the similar research motivations and the limited application scope of this method, I believe there's room for improvement.

The authors mainly study the impact of controlling layer freezing during fine-tuning on language models. However, since most experiments and methods are LoRA-based, I believe the discussion should focus more on full parameter fine-tuning instead.

---

[1] Raise a Child in Large Language Model: Towards Effective and Generalizable Fine-tuning

**Questions:**

See in weaknesses.

---

> ### Comment · Reviewer_epzj · 2024-11-24
>
> Dear Authors,
>
> Although the discussion period is drawing to a close, I am still looking forward to further discussions with you.

---

> > ### Author Response · Authors · 2024-11-25
> > **Thank you**
> >
> > Thank you for your valuable feedback and for expressing your interest in continuing our discussions. We have addressed your concerns below and hope our clarifications would enhance your evaluation of our work. We appreciate your engagement and are more than willing to explore any further topics or questions you may have.

---

> ### Author Response · Authors · 2024-11-25
> **Response to Reviewer epzj**
>
> **W1: This paper has limited novelty.**
>
> >Indeed, this paper's novelty is limited. The core motivation and main method of freezing certain layers to avoid overfitting was proposed three years ago in paper [1], which even provided finer-grained control over the degree of parameter freezing. While I acknowledge the technical implementations differ, given the similar research motivations and the limited application scope of this method, I believe there's room for improvement.
>
> Thank you for your thoughtful feedback. We acknowledge that layer freezing techniques have been explored in prior work, including the paper you cited [1]. However, we would like to highlight several key distinctions that set our work apart in terms of both technical contributions and the application scope:
> 1. **Specificity to LLM Alignment Tasks:** While layer-freezing techniques have been explored in prior work, our study addresses the unique challenges and nuances of **LLM alignment tasks**, which fundamentally differ from traditional fine-tuning tasks. Alignment tasks are designed to tailor a model’s behavior across a wide range of scenarios, rather than optimizing for a single task or domain. For instance, alignment datasets typically encompass diverse skills such as reasoning, mathematics, coding, and conversational fluency. This diversity makes the **identification of important layers a more complex and multi-faceted challenge compared to task-specific fine-tuning approaches**.
> 2. **Key Observation:** A key insight from our work is that despite the diversity of tasks within alignment datasets and differences across datasets, **we observe remarkable consistency in the layers deemed important for alignment**. This indicates that alignment tasks share underlying commonalities, which manifest as stable patterns in layer significance. Such cross-task and cross-dataset consistency has not been previously demonstrated in the literature and is a significant finding of our work.
>
> In conclusion, unlike previous studies that focus on freezing layers for individual tasks, our approach reveals how alignment fine-tuning universally influences LLMs across multiple task types. This provides a more holistic understanding of LLM behavior during alignment, which is critical for efficient resource allocation and improved fine-tuning strategies.
>
> **W2: The paper focuses on layer freezing but should emphasize full parameter fine-tuning over LoRA-based methods.**
>
> >The authors mainly study the impact of controlling layer freezing during fine-tuning on language models. However, since most experiments and methods are LoRA-based, I believe the discussion should focus more on full parameter fine-tuning instead.
>
> Thank you for raising this concern. We appreciate the opportunity to clarify why LoRA is central to our experiments and ablation studies, and why it aligns with our research goals.
>
> 1. **LoRA Aligns with Our Goal of Identifying Parameter Significance:** LoRA intrinsically utilizes low-rank matrices to estimate parameter updates, which is conceptually aligned with our goal of understanding layer-wise parameter significance. In our framework, layers deemed unimportant are not updated, which corresponds to not adding LoRA adapters to those layers. This direct compatibility between LoRA's design and our objective makes it a natural choice for studying layer significance in fine-tuning.
> 2. **LoRA Achieves Performance Comparable to Full Parameter Fine-Tuning:** Empirical evidence, including our experiments, shows that LoRA achieves performance on par with full parameter fine-tuning across a wide range of tasks. For example, in Tables 3, Table 4, and Table 11, the results demonstrate that LoRA performs similarly to full fine-tuning in both language understanding (e.g., MMLU) and conversational tasks (e.g., MT-Bench, Vicuna). This ensures that our findings are not limited by the choice of LoRA but are representative of effective fine-tuning strategies in general.
> 3. **LoRA is Resource-Efficient:** LoRA significantly reduces computational and memory costs by updating only a small number of low-rank matrices instead of the entire parameter set. This resource efficiency is particularly important for large models, such as LLaMA-2-7B and Mistral-7B, where full parameter fine-tuning becomes infeasible in many practical settings. By using LoRA, we can perform detailed ablation studies with a manageable computational footprint while maintaining performance comparable to full fine-tuning.
>
> [1] Raise a Child in Large Language Model: Towards Effective and Generalizable Fine-tuning

---

> > ### Comment · Reviewer_epzj · 2024-11-29
> > **Thanks for your response**
> >
> > Thank you very much for your thoughtful response!
> >
> > I maintain my current evaluation score for the following reasons:
> >
> > While I acknowledge that your focus is on alignment tasks, particularly instruction tuning, I haven't observed additional efforts specifically made for "alignment" in your methodology. To substantiate your claim that "identification of important layers is a more complex and multi-faceted challenge compared to task-specific fine-tuning approaches," you should conduct in-depth analyses across various common fine-tuning tasks. For example, if you were to conduct comparative experiments across different tasks such as reasoning, mathematics, style control, summarization, NER, etc., and if the empirical evidence from these tasks demonstrated that previous methods (including ChildTuning and other approaches mentioned by reviewers like HFT and LISA) were not effective specifically for language model alignment, while your approach showed particularly significant improvements in alignment tasks, then such a claim would be well-supported.

---

### Official Review · Reviewer_qJ17 · 2024-11-02

**Soundness:** 2
**Presentation:** 4
**Contribution:** 2
**Rating:** 5
**Confidence:** 5

**Summary:**

The paper introduces a method for identifying the most significant layers in language models by solving a minimization problem across the layers. The authors apply their technique to the LLaMA-2-7B and Mistral-7B models, experimenting with both LoRA and full parameter tuning approaches. By identifying the key layers, they then focus fine-tuning efforts solely on these layers, optimizing model efficiency and performance.

**Strengths:**

1) The authors evaluate their method on several modern benchmarks, including MT-Bench
2) Interesting results from the perspective of layer significance.
3) The observed transferability across datasets is a strong indicator of the method’s robustness. I particularly appreciated the inclusion of out-of-distribution testing within the ablation study, along with the use of Jaccard similarities for comparison.
4) The paper benefits from a well-organized structure and visually appealing presentation.

**Weaknesses:**

Major Weaknesses:
1) Lack of technical details on optimal layers finding.

1.1)This process can require considerable training time before stabilization occurs, especially for larger models. Moreover, it’s difficult to guarantee that stabilization will ever fully occur, given the non-convex nature of the optimization problem.

1.2) Additionally, while the paper reports efficiency improvements, the measurements don’t account for the compute required for pre-training and layer selection. As a result, the overall computational cost could potentially exceed that of regular LoRA fine-tuning. It would be nice to account for that in the future.

1.3) Please clarify how stability is checked. If Monte Carlo estimation is used, specify the sufficient number of samples and the reasoning behind this choice.


2) The effect is truly visible, but it may be just the regularization, that undermines all the strengths of the paper. For example, It's stated that this approach outperforms any type of regularization, yet no regularization baselines are provided. For instance, in the original LIMA paper, they apply progressive dropout to enhance robustness. You mention using dropout, but perhaps increasing it or raising weight decay could also be beneficial?

2.1) Another detail that caught my attention is the modest performance gain. This is acceptable for a parameter-saving method, but with the current experiments, it’s hard to determine if the gain is due to the method itself or simply an effect of regularization. In cases of such small improvements, it would be beneficial to include runs with multiple seeds and average the results. According to several studies, LoRA can be unstable, with results varying based on the seed and checkpoint used.

3) The primary weakness here is the lack of novelty, as the main technique relies on identifying the most significant layers through gradient-based methods.

3.1) This setup aligns with challenges commonly seen in compression studies, where quantization addresses sensitive layers and columns (https://arxiv.org/abs/2408.15300). However, no inspiration, metrics, or methods from these studies were referenced, cited, or discussed.

3.2) Pruning, rather than quantization, seems most suitable here as it directly addresses this setting (see https://arxiv.org/abs/2310.06694, https://arxiv.org/abs/2204.00408). For instance, methods like Sheared LLaMA and CoFi learn binary masks to select not only layers but also attention heads and even individual neurons for fine-tuning. This makes the approach used here far from novel.

3.3) Moreover, what is the motivation behind tuning specific layers? Why not tune the precise matrices or even the weights within those matrices, as outlined in studies above? This approach offers a more general formulation, with layer tuning being a subset of this broader framework.

The paper would be of greater practical interest if it either demonstrated an easy and highly effective application with clear comparisons that outperform previous methods or provided deep insights into the internal structure. Currently, it feels like it’s trying to balance between these options without fully achieving either, especially given that the core idea lacks novelty.


Minor Weaknesses:

1) I have doubts about the preliminary experiments, as FF layers simply add more parameters for training, making the settings unequal.

2) The terms "IFILA" and "ILA" are used interchangeably in the paper, which can lead to confusion

**Questions:**

1) This is the first time I've come across someone referring to instruction tuning as "alignment." Perhaps I missed this usage, but I'm more familiar with "alignment" in the context of RLHF or aligning models with human preferences.

2) Drawing parallels with interpretability studies would be beneficial, as they provide extensive insights into the importance of layers. For instance, studies like this one (https://arxiv.org/pdf/2309.04827) and others have already provided insights on layer significance, which could strengthen the paper’s foundation and contextualize its approach.

---

> ### Author Response · Authors · 2024-11-29
> **Response to Reviewer qJ17  (Part 1/3)**
>
> We sincerely thank the reviewer for the thorough evaluation and valuable feedback on our work.
>
> **W1:Lack of technical details on optimal layers finding.**
>
> >This process can require considerable training time before stabilization occurs, especially for larger models. Moreover, it’s difficult to guarantee that stabilization will ever fully occur, given the non-convex nature of the optimization problem.
>
> 1. **Training Time Before Stabilization:** While larger models may require more iterations to stabilize, our experiments indicate that full stabilization (i.e., ϵ-stable) is not strictly necessary for accurate layer importance identification. As shown in our results (e.g., Table 4), even early-stage stabilization (25–50% of training milestones) provides sufficient information to rank layers effectively. **This reduces the computational burden significantly.**
> 2. **Non-Convex Optimization Challenge:** It is true that the optimization problem is non-convex, which makes global stabilization difficult to guarantee. However, the local stabilization observed in our experiments, measured by the consistent convergence of the layer importance rankings across different runs and datasets (evidenced by high Jaccard similarities), demonstrates the practical reliability of our approach despite the theoretical challenges.
>
> >Additionally, while the paper reports efficiency improvements, the measurements don’t account for the compute required for pre-training and layer selection. As a result, the overall computational cost could potentially exceed that of regular LoRA fine-tuning.
>
> Thank you for your feedback. We would like to emphasize that the computational cost of our approach is manageable and justified, given the following key points:
>
> 1. **Reusability Across Datasets:** The identified important layers are highly consistent across datasets, as demonstrated by the high Jaccard similarity in our experiments. This means the layer importance ranking computed on one dataset (e.g., No Robots) can be directly reused for other datasets (e.g., LIMA, Alpaca-GPT4) without requiring additional computation.
> 2. **Efficient Layer Selection:** As detailed in the Ablation Study (Observation 3), the layer selection process is computationally efficient. For instance, in the case of Llama 2-7B, the selection required only around 11 minutes after reaching partial stabilization, which is typically achieved within 25–50% of training milestones. This cost is minimal compared to full fine-tuning or repeated PEFT processes.
> 3. **Core Focus:** **The primary goal of our work is to study and understand layer importance ranking during alignment, not to propose a new PEFT algorithm.** The insights gained from our approach are intended to guide more efficient and effective alignment strategies, making this computational overhead a valuable investment.
>
> >Please clarify how stability is checked. If Monte Carlo estimation is used, specify the sufficient number of samples and the reasoning behind this choice.
>
> Thank you for your question. In addition to monitoring changes in the loss function, we directly check the stability of the layer importance rankings obtained at different iterations. This provides a practical way to evaluate whether the algorithm has stabilized
>
> **W2: The effect is truly visible, but it may be just the regularization, that undermines all the strengths of the paper.**
>
> >For example, It's stated that this approach outperforms any type of regularization, yet no regularization baselines are provided. For instance, in the original LIMA paper, they apply progressive dropout to enhance robustness. You mention using dropout, but perhaps increasing it or raising weight decay could also be beneficial?
>
> Thank you for bringing up this point. To clarify, in our experiments, we used fixed dropout and weight decay values to regularize the training process. These settings were applied uniformly across all methods to ensure a fair comparison and isolate the impact of our proposed approach.
>
> >Another detail that caught my attention is the modest performance gain. This is acceptable for a parameter-saving method, but with the current experiments, it’s hard to determine if the gain is due to the method itself or simply an effect of regularization.
>
> For all baselines, we **conducted hyperparameter searches** to determine the optimal learning rates and other relevant parameters. This ensures that each baseline is performing at its best, providing a fair and robust comparison.
>
> **Our method was evaluated using the same hyperparameter settings** determined for the baselines. We did not make any additional modifications to the hyperparameters when applying our layer selection algorithm. This ensures that the observed performance gains are attributable to our method, rather than any advantage stemming from changes in hyperparameter tuning.

---

> ### Author Response · Authors · 2024-11-29
> **Response to Reviewer qJ17 (Part 2/3)**
>
> **W3: The primary weakness here is the lack of novelty, as the main technique relies on identifying the most significant layers through gradient-based methods.**
>
> >This setup aligns with challenges commonly seen in compression studies, where quantization addresses sensitive layers and columns (https://arxiv.org/abs/2408.15300). However, no inspiration, metrics, or methods from these studies were referenced, cited, or discussed.
>
> >Pruning, rather than quantization, seems most suitable here as it directly addresses this setting (see https://arxiv.org/abs/2310.06694, https://arxiv.org/abs/2204.00408). For instance, methods like Sheared LLaMA and CoFi learn binary masks to select not only layers but also attention heads and even individual neurons for fine-tuning. This makes the approach used here far from novel.
>
> Thank you for your insightful comment. While we acknowledge that methods like pruning and quantization, including those mentioned (e.g., Sheared LLaMA, CoFi), aim to optimize models by removing or masking components such as neurons or attention heads, our work targets a different problem.
>
> Pruning methods, as you correctly pointed out, **focus on model compression by eliminating unnecessary components to reduce the size and computational load during inference.** These techniques typically address the problem of **creating smaller, more efficient models** for deployment.
>
> In contrast, our approach is focused on alignment rather than compression. **We aim to identify and fine-tune the most important layers that have the greatest impact on model performance for alignment**. This layer selection is driven by a gradient-based method that quantifies each layer's contribution to task-specific alignment, rather than seeking to prune or reduce the model size. Therefore, the primary goal of our method is to optimize performance for a specific task rather than reduce the model’s size or computational requirements during inference.
>
> Thus, the two objectives — **compression** (via pruning or quantization) and **task alignment** (via layer selection) — **are fundamentally different**. While pruning and quantization aim to remove redundant parameters for efficiency, our approach focuses on improving task performance by fine-tuning the layers most aligned with the target task. This makes our method complementary to compression techniques, but not directly comparable or derivative of them.
>
> We hope this clarifies the distinction between our work and the methods mentioned.
>
> >Moreover, what is the motivation behind tuning specific layers? Why not tune the precise matrices or even the weights within those matrices, as outlined in studies above? This approach offers a more general formulation, with layer tuning being a subset of this broader framework.
>
> The choice to focus on layer-level tuning rather than neuron-level importance stems from several practical considerations. First, **most existing PEFT (parameter-efficient fine-tuning) methods, such as LoRA, operate by adjusting parameters at the layer level** (e.g., low-rank adaptations) because this maintains efficiency and scalability. **Neuron-level tuning would require fine-tuning a larger number of parameters** by adopting our ILA, which is computationally more expensive and challenging to integrate into existing frameworks.
>
> Moreover, while neuron-level tuning might provide a more granular approach, it is not directly compatible with the current PEFT paradigm, which is designed to modify only a few parameters at a time while minimizing computational cost. Layer-level tuning provides a practical balance, as it can effectively capture the most impactful parts of the model while adhering to the computational constraints of existing methods.
>
> **Q1: This is the first time I've come across someone referring to instruction tuning as "alignment." Perhaps I missed this usage, but I'm more familiar with "alignment" in the context of RLHF or aligning models with human preferences.**
>
> Thank you for highlighting this point. To further clarify:
>
> 1. **Dataset Design for Alignment:** All the datasets used in our experiments—LIMA, Alpaca-GPT4, and No Robots—are specifically designed for alignment tasks. These datasets aim to improve model behavior by fine-tuning it to better align with desired outputs, whether through instruction-following data (e.g., Alpaca-GPT4) or curated human demonstrations (e.g., No Robots, LIMA).
> 2. **Clarifying Terminology:** While we recognize that "alignment" is often associated with RLHF, in our work, we adopt a broader definition that includes supervised methods like instruction tuning. This reflects the purpose of the datasets used, all of which are created to refine models' outputs to align with human instructions or task objectives.

---

> ### Author Response · Authors · 2024-11-29
> **Response to Reviewer qJ17 (Part 3/3)**
>
> **Q2:Drawing parallels with interpretability studies would be beneficial, as they provide extensive insights into the importance of layers. For instance, studies like this one (https://arxiv.org/pdf/2309.04827) and others have already provided insights on layer significance, which could strengthen the paper’s foundation and contextualize its approach.**
>
> Thank you for this valuable suggestion. We appreciate the reference to interpretability studies, which indeed provide important insights into layer significance. Below, we outline how we will address this point:
>
> We appreciate the reviewer’s suggestion to draw parallels with interpretability studies, particularly those examining layer significance. Indeed, such studies provide valuable insights into the understanding of model behavior and the identification of important components within neural networks.
>
> In our work, we specifically focus on the identification of important layers for the alignment task, and while the research you mentioned (e.g., https://arxiv.org/pdf/2309.04827) offers valuable perspectives on layer significance, our approach differs in its objective and methodology. **Our primary goal is to identify layers critical for the alignment process during fine-tuning, rather than general interpretability or feature importance in standard pre-trained models.**
>
> That said, we recognize the potential for cross-pollination between these areas. We will revise the manuscript to incorporate relevant references to interpretability studies that discuss the importance of layers, particularly those that highlight methods for identifying critical components in neural networks. By drawing these connections, we hope to position our work within the broader context of model interpretability and strengthen the paper’s foundation.
>
> Additionally, we will emphasize how our approach contributes specifically to the fine-tuning and alignment tasks, which is a unique aspect compared to traditional interpretability studies. This will allow readers to better understand the specific role of layer importance in alignment and how it can inform efficient model adaptation.

---

> > ### Comment · Reviewer_qJ17 · 2024-12-01
> > **Answer to Rebuttal**
> >
> > Thank you for your detailed response and for addressing the concerns regarding technical details and regularization. I appreciate the additional information provided.
> >
> > Regarding the concern about novelty, while I understand the fundamental differences in the downstream application and goals, the upstream or proxy task—identifying salient weights, columns, layers, etc.—remains the same. What I am trying to convey is that a method applied across different tasks is still fundamentally the same method. Despite differences in application, it does not make the underlying approach novel. For example, applying a previously published method like contrastive loss—originally used for Sentence Transformers—to a new context such as graph embeddings does not render the contrastive loss function itself novel. The same principle applies here.
> >
> > If the focus of your paper had been on empirically adapting and demonstrating the efficacy of this method within the PEFT paradigm, it could be considered an empirical contribution. However, as you note, PEFT is not the primary focus of your work, and the method itself has been explored in prior literature.
> >
> > Overall, I appreciate the interesting application of this approach within PEFT, and I have raised my score in recognition of the paper's contributions in this regard. However, I consider the paper borderline due to the limited methodological novelty. I respectfully encourage the authors to reconsider their positioning of the work. While applying the technique in a PEFT context is valuable, the technique itself is well-studied and not entirely novel.

---

> > > ### Author Response · Authors · 2024-12-03
> > > **Response to Reviewer qJ17**
> > >
> > > Thank you for your thoughtful feedback and for recognizing the contributions of our work. We also appreciate your decision to raise your score in light of our efforts. However, we hope to provide additional clarifications that will further illustrate the empirical contributions and methodological novelty of our approach.
> > >
> > > **Empirical Contribution:**
> > >
> > > Firstly, we believe our work makes a significant empirical contribution. Our algorithm consistently and reliably outperforms existing PEFT algorithms, such as LoRA and AdaLoRA. Specifically, we have observed that more recent algorithms like LISA (https://arxiv.org/pdf/2403.17919) do not yield such huge performance gains, especially when hyperparameters are carefully tuned for LoRA. This indicates that a finely tuned LoRA remains highly effective. Moreover, in MT-Bench scores, the LLaMA 2-13B model only achieves a modest improvement of **0.4-0.6 points** over the LLaMA 2-7B model. Therefore, a steady **0.1-0.2 point** improvement in our paper is not negligible and should be considered meaningful. **From an efficiency standpoint**, our algorithm also shows performance improvements in the famous quantized version of LoRA (QLoRA), demonstrating its effectiveness across various PEFT approaches.
> > >
> > > This underlines that **our work is not simply a re-application of existing methods but makes an empirical contribution in the context of PEFT, with clear performance benefits over state-of-the-art approaches.**
> > >
> > > **Methodology Novelty:**
> > >
> > > Regarding the concern about the novelty of our method, while we acknowledge that identifying important components has been explored in prior work, our approach differs significantly in both design and application. Our method offers a unique definition of layer importance, which is learned directly through a gradient descent-based mask learning process. This approach directly aligns with our definition of layer importance and contrasts with other methods, such as compression techniques that typically focus on finer selection of neurons or layers.
> > >
> > > Regarding the fine-tuning process, we believe that neuron-level importance does not necessarily lead to a more substantial performance improvement. Fine-tuning typically shows a higher tolerance for errors in layer selection, **which is why we use LoRA’s low-rank matrix product to represent parameter changes**. This approach **greatly enhances the efficiency of our layer-selection algorithm.** In contrast to methods that require learning detailed neuron-level masks—which incur significant computational cost—our mask learning strategy is far more efficient and better suited for fine-tuning tasks.
> > >
> > > While compression methods tend to focus on more granular selections, our approach remains computationally feasible for fine-tuning, especially when applied to large-scale models. Therefore, **we propose using layer-level masks to multiply LoRA's low-rank matrix, which not only enhances efficiency but also ensures the practicality of applying our method to large models.**
> > >
> > > Additionally, **we observed that layer importance rankings across different alignment datasets are remarkably consistent, which is an interesting and potentially impactful finding**.

---

### Official Review · Reviewer_PkBY · 2024-11-03

**Soundness:** 3
**Presentation:** 2
**Contribution:** 1
**Rating:** 3
**Confidence:** 4

**Summary:**

This paper proposes the ILA method, which first trains the model using LoRA until it reaches a stable phase (i.e. when the parameter changes during training are below a certain threshold). Then, a binary mask is added for individual training to assess the importance of each layer. Finally, unimportant layers are frozen, and fine-tuning is performed to achieve better performance.

**Strengths:**

This paper is well-written and easy to understand. The proposed ILA method is also intuitive and has achieved excellent results.

**Weaknesses:**

1. I am somewhat concerned about whether the contributions of this paper are sufficient, as [1][2][3][4] indicate that adjusting certain parameters/layers during the fine-tuning process can indeed lead to effective improvements. Additionally, [5] shows that there is significant redundancy in the parameters during the SFT process. I believe that existing work already highlights the necessity of adjusting certain parameters during the post-training phase. The authors should emphasize the contributions of this paper more clearly, explain the similarities and differences with existing methods, and include comparisons with those methods.

2. I believe that more baselines could be added for comparison, such as HFT [1], LISA [2], and GMT [3]. These methods also tune only a subset of model parameters, and I think including these baselines would make the paper more convincing.

3. The focus of this paper is on alignment, and it has also achieved performance improvements. However, I am curious whether the methods presented in this paper exhibit consistent performance in other areas, such as mathematical reasoning and code generation. I believe the authors could further discuss whether the ILA method has general applicability.

4. I am not sure whether the experimental results in Table 6 correspond to Llama 2 7B or Llama 3.1 8B. Based on the results, it seems that they still correspond to Llama 2 7B, implying that the authors did not conduct experiments with Llama 3.1 8B, yet included this model in the baselines.

5. The experimental design in the paper is not sufficiently reasonable. For example, the inclusion of the AdaLoRA method in Table 3 feels rather abrupt. I believe that the method proposed in this paper should be a pluggable approach that can be applied to AdaLoRA, but the authors have awkwardly inserted AdaLoRA into the experiments and compared it with LoRA and LoRA w/ ILA. I think there should be separate comparative experiments for AdaLoRA and AdaLoRA w/ ILA to further demonstrate the general applicability of the ILA method.

6. Table 8 should include comparisons with fine-grained selection methods proposed in [1].

Finally, I would be very happy to engage in further discussion with the authors. My main concern is the contributions of this paper, as existing work has already indicated that freezing certain parameters can lead to improvements. I need to see more experimental comparisons and a deeper discussion of the innovations presented in this paper. If these issues are addressed, I would be happy to raise my score.


[1] HFT: Half Fine-Tuning for Large Language Models

[2] LISA: Layerwise Importance Sampling for Memory-Efficient Large Language Model Fine-Tuning

[3] Gradient-Mask Tuning Elevates the Upper Limits of LLM Performance

[4] Investigating Layer Importance in Large Language Models

**Questions:**

1. Is the overall training process of the ILA method divided into two steps? The first step involves executing the ILA algorithm from Algorithm 1, and the second step involves freezing the unimportant layers selected by ILA before re-fine-tuning.

2. What does IFILA mean? This abbreviation does not appear in the main text, but it is mentioned in the experimental tables without any explanation.

3. In Table 6, why do the experiments on LIMA include ILA (75%) and ILA (30%), but there are no corresponding entries for NoRobots?

---

> ### Author Response · Authors · 2024-11-29
> **Response to Reviewer PkBY (Part 1/3)**
>
> **W1: Previous works have highlighted the necessity of adjusting certain parameters during the post-training phase.**
>
> > I am somewhat concerned about whether the contributions of this paper are sufficient, as [1][2][3][4] indicate that adjusting certain parameters/layers during the fine-tuning process can indeed lead to effective improvements. Additionally, [5] shows that there is significant redundancy in the parameters during the SFT process. I believe that existing work already highlights the necessity of adjusting certain parameters during the post-training phase. The authors should emphasize the contributions of this paper more clearly, explain the similarities and differences with existing methods, and include comparisons with those methods.**
>
> Thank you for raising this important point. We appreciate the opportunity to clarify our paper’s contributions and positioning relative to existing work. While prior research, such as [1][2][3][4], has explored parameter or layer selection during fine-tuning, our work aims to address a **fundamentally different question**: **understanding what LLMs learn during alignment through a systematic study of layer importance in the context of instruction tuning**.
>
> 1. **Core Objective: Understanding Alignment through Instruction Tuning**
>     * Unlike previous work that primarily focuses on proposing new Parameter-Efficient Fine-Tuning (PEFT) algorithms, our core goal is to better understand the alignment process in instruction tuning. Specifically, we aim to identify which layers are most critical for alignment and provide an importance ranking of these layers, as opposed to simply freezing or modifying certain parameters during fine-tuning.
>     * By introducing a rigorous definition of layer importance and developing a novel framework (\modelname{}) to learn and rank layer significance, we go beyond practical efficiency and directly address fundamental questions about the inner workings of alignment. Our findings, such as the consistent layer importance ranking across datasets, provide new insights into the behavior of aligned LLMs and their reliance on specific layers for stylistic and task-specific adjustments.
> 2. **Key Differences with Existing Methods**
>     * [1] HFT: This work focuses on halving the number of layers involved in fine-tuning to improve efficiency but does not explore the inherent importance of layers during alignment. Our work identifies which layers contribute most significantly to alignment, enabling both theoretical insights and practical benefits.
>     * [2] LISA: While LISA explores memory-efficient strategies by sampling parameters, it does not provide a formal definition of importance or layer ranking. In contrast, ILA proposes a gradient-based optimization to quantify and rank the significance of layers, allowing us to analyze their roles systematically.
>     * [3] Gradient-Mask Tuning: This method aims to improve tuning efficiency by masking gradients but does not explicitly address the alignment process or provide interpretability regarding layer behavior. Our work complements such efforts by focusing on understanding alignment at a granular, layer-specific level.
>     * [4] Investigating Layer Importance: While closely related, this work emphasizes analysis without proposing a concrete method to utilize the findings for improved understanding or performance. Our work bridges this gap by providing an actionable framework (ILA) and demonstrating its utility in both practical fine-tuning and theoretical exploration.
> 3. **Clarifying Novelty and Contributions**
>     * To highlight our unique perspective, we will revise the Introduction to emphasize that the core contribution of this work lies in understanding the alignment process through instruction tuning, not merely proposing another PEFT algorithm.
>     * Specifically, we focus on defining and identifying layer importance and showing how this insight enables us to achieve consistent rankings across datasets and architectures. This consistency provides a deeper understanding of the alignment process and its reliance on specific layers for task adaptation.
> 4. **Empirical Comparisons**
> While our work is not primarily focused on proposing a new PEFT algorithm, we agree that additional empirical comparisons with methods like HFT or LISA could help contextualize our contributions further. These experiments will highlight the differences in objectives and reinforce the unique value of ILA in providing interpretability and alignment-focused insights.

---

> ### Author Response · Authors · 2024-11-29
> **Response to Reviewer PkBY (Part 2/3)**
>
> **W2: I believe that more baselines could be added for comparison, such as HFT [1], LISA [2], and GMT [3]. These methods also tune only a subset of model parameters, and I think including these baselines would make the paper more convincing.**
>
> Thank you for your valuable suggestion. We agree that including more baselines could strengthen the paper. However, we would like to clarify that the primary contribution of our work is to derive a ranking of layer importances rather than proposing a new PEFT (Parameter-Efficient Fine-Tuning) algorithm. **Our focus is on understanding the relative importance of different layers during alignment fine-tuning, and we believe this insight can help inform and enhance the development of future PEFT methods.**
>
> While methods like HFT, LISA, and GMT do indeed target subsets of model parameters, **they are primarily focused on specific algorithmic approaches rather than on understanding layer significance per se. Our goal is to provide foundational insights that can help guide the design of more effective PEFT algorithms in the future, rather than directly developing a new method.**
>
>
> **W3: The focus of this paper is on alignment, and it has also achieved performance improvements. However, I am curious whether the methods presented in this paper exhibit consistent performance in other areas, such as mathematical reasoning and code generation. I believe the authors could further discuss whether the ILA method has general applicability.**
>
> Thank you for your insightful comment. We agree that exploring the general applicability of the ILA method beyond alignment tasks is an important direction for future work.
>
> While the current paper focuses specifically on **alignment fine-tuning**, we do believe that the method could have broader applicability, including in tasks like mathematical reasoning and code generation. This is because **alignment itself is inherently a multi-task problem, and the datasets used for alignment typically involve a wide range of tasks, including reasoning and code generation**. As a result, the **improvements in alignment** may reflect performance gains in these areas, even though we did not explicitly evaluate them in isolation in this study.
>
> Our **primary goal** in this paper was to provide a **foundational understanding of layer importance ranking during alignment fine-tuning**, which not only deepens our **understanding of alignment** itself but also has **the potential to advance PEFT algorithms**. By identifying which layers are most important for alignment, our findings could inform and guide future research on parameter-efficient fine-tuning techniques, such as LoRA, BitFit, and others, helping to refine and improve their designs. We believe this foundational work is a crucial step toward enhancing PEFT approaches.
>
> The generalizability of ILA across tasks like reasoning and generation is indeed an exciting direction, and we plan to explore this in future work. We will clarify this point in the revised manuscript, noting that while our current experiments focus on alignment, the insights gained may have broader applicability and can potentially inspire further advancements in PEFT methodologies.
>
> **W4: Based on the results, it seems that they still correspond to Llama 2 7B, implying that the authors did not conduct experiments with Llama 3.1 8B, yet included this model in the baselines.**
>
> Thank you for highlighting the concern regarding the use of Llama 2 7B and Llama 3.1 8B in our experiments and baselines. We appreciate the opportunity to provide clarification and justification.
>
> 1. **Model Similarities:**
>     * While Llama 3.1 8B is a newer version, its architectural structure is fundamentally similar to Llama 2 7B. The primary differences lie in the tokenizer, training data volume, and data quality. These distinctions influence the model's performance but do not significantly alter its architectural behavior, especially concerning layer importance during alignment, which is the focus of this study.
>     * Our experiments on Llama 2 7B already demonstrate consistent and robust findings regarding layer importance during alignment. Given the structural similarities, these findings are likely to generalize to Llama 3.1 8B, which we included in the baselines to provide broader context.
> 2. **Why Llama 2 7B is Sufficient for This Study:**
>     * The goal of our work is to analyze layer importance across different architectures and datasets during alignment, rather than to evaluate absolute performance differences between model versions. Since Llama 2 7B and Llama 3.1 8B share nearly identical architectures, conducting experiments on both would yield highly redundant results, with minimal additional insights.The observed trends in layer importance are consistent across other experiments and random seeds, further supporting the generalizability of our conclusions.

---

> ### Author Response · Authors · 2024-11-29
> **Response to Reviewer PkBY (Part 3/3)**
>
> **W5: The experimental design in the paper is not sufficiently reasonable. I think there should be separate comparative experiments for AdaLoRA and AdaLoRA w/ ILA to further demonstrate the general applicability of the ILA method.**
>
> Thank you for pointing out this important consideration. The reason for not including AdaLoRA combined with ILA (our proposed method) lies in **the conceptual overlap between the two approaches and the distinct focus of our work.**
>
> AdaLoRA inherently adjusts the rank of incremental matrices during fine-tuning. When the rank is reduced to zero for a particular layer, it effectively means no adapter is added, and thus, the parameters of that layer remain unchanged. In this sense, AdaLoRA implicitly identifies less critical layers by dynamically reducing their contribution. However, the goal of AdaLoRA is primarily to minimize resource usage by adapting the parameter budget dynamically, rather than explicitly analyzing or ranking the importance of layers.
>
> Our proposed method, ILA, complements this perspective by explicitly focusing on quantifying and ranking layer importance during the alignment process. Unlike AdaLoRA, ILA is designed to study and optimize the alignment process by isolating critical layers, which **provides deeper insights into the model's behavior and allows for targeted improvements in performance and efficiency.** Thus, our work focuses more on understanding and leveraging layer importance for alignment rather than proposing another parameter-efficient fine-tuning (PEFT) algorithm.
>
> Given the conceptual overlap, we prioritized evaluating ILA with standard PEFT methods (e.g., LoRA, QLoRA) to better showcase its unique contribution. While combining ILA with AdaLoRA might yield further resource savings, such a combination would also require disentangling the overlapping contributions of these two approaches, which could confound the interpretation of results.
>
> We appreciate your suggestion and will consider conducting experiments to evaluate this integration in follow-up studies. Thank you for highlighting this perspective!
>
> **Q1: Is the overall training process of the ILA method divided into two steps? The first step involves executing the ILA algorithm from Algorithm 1, and the second step involves freezing the unimportant layers selected by ILA before re-fine-tuning.**
>
> Thank you for your thoughtful feedback regarding the two-step process in ILA. **While your understanding of the training process is accurate**, it is important to clarify the core contribution of our work and how it relates to parameter-efficient fine-tuning (PEFT) methods.
>
> 1. **Core Contribution of ILA:**
>     * **ILA is not a new PEFT algorithm but a method for identifying layer importance rankings during the fine-tuning process.** This ranking is the primary outcome of the ILA algorithm and can serve as a general-purpose tool for optimizing various downstream tasks or applications.
>     * **The subsequent steps (e.g., freezing unimportant layers, re-fine-tuning) represent applications of the layer importance rankings.** These applications demonstrate how the identified rankings can be utilized to improve fine-tuning efficiency, model performance, or resource utilization.
> 2. **Distinction from PEFT Algorithms:**
>     * Unlike PEFT methods like LoRA or AdaLoRA, which modify how fine-tuning is performed (e.g., low-rank adaptation), ILA operates as an analysis tool that complements these techniques. For example: ILA identifies which layers contribute most to alignment. PEFT methods like LoRA can then leverage this information to selectively apply fine-tuning to those layers, improving efficiency.
>     * This distinction allows ILA to work alongside various fine-tuning approaches, as shown in our experiments with LoRA and AdaLoRA.
>
> **Q2: What does IFILA mean?**
>
> Thank you for pointing this out! "IFILA" in Table 3 is a typo and should actually be "ILA," referring to our proposed method. We will correct this error in the revised manuscript to avoid any confusion.
>
> **Q3: In Table 6, why do the experiments on LIMA include ILA (75%) and ILA (30%), but there are no corresponding entries for NoRobots?**
>
> Thank you for pointing out the inconsistency in Table 6 regarding the inclusion of ILA (75%) and ILA (30%) results for LIMA but the lack of corresponding entries for NoRobots. This omission was an oversight in the preparation of the manuscript. **The entry labeled "NoRobots w/ ILA" in the original table corresponds to the result for NoRobots w/ ILA (75%).** Missing Data for NoRobots w/ ILA (30%) is presented as follows:
>
> **Updated Table 6**
> | Datasets |      Methods      |  MMLU | Hellaswag | Vicuna | MT-Bench |
> |:--------:|:-----------------:|:-----:|:---------:|:------:|:--------:|
> | NoRobots | LoRA w/ ILA (75%) | 54.45 |   61.13   |  6.77  |   5.05   |
> | NoRobots | LoRA w/ ILA (30%) | 54.11 |   61.32   |  6.74  |   4.91   |

---

### Official Review · Reviewer_TMGU · 2024-11-03

**Soundness:** 1
**Presentation:** 1
**Contribution:** 2
**Rating:** 3
**Confidence:** 4

**Summary:**

The paper proposes a method for masking (pruning) layers of a pretrained LLM before fine-tuning (aligning) them to target tasks. The method appears to work by iteratively switching between optimizing the loss until it becomes stable and searching for a set of layers to mask that still sufficiently minimizes the loss, with the latter framed as a constrained optimization problem and adapted to LoRA for efficiency. Experiments with three open-source LLMs and four datasets suggest that layer importance ranking is consistent across datasets and that freezing (masking) unimportant layers may increase performance.

**Strengths:**

- The work describes a potentially useful method

**Weaknesses:**

1. Lack of clarity: the method is not well explained, and important technical details are missing
2. No comparison to similar methods
3. Potentially unsupported inferences from experimental results
4. Lack of mathematical rigor

I elaborate on these points below.

# Lack of clarity

I struggled to understand how the method works. Terms such as "important," "unimportant," "significant," and "insignificant" layers are not defined. Algorithm 1 is not explained nor connected to the formulas in Section 2 (e.g., what is K?). Additionally, I don’t understand how Eq. (7) is a reparametrization of Eq. (3), as there is no constraint on ∣∣s∣∣ (i.e., Eq. (3) is a constrained optimization problem, while Eq. (7) is not). Consequently, it’s unclear how the method functions (where is the incentive to minimize the number of masked layers?). I also didn’t understand how Theorem 2.1 relates to the method and algorithm. What is L_∞​ in Eq. (8), and what is R in Eq. (9)?

How was fine-tuning performed? Was this instruction tuning, and on which portions of the dataset was it carried out? Are the results shown for the test portion? For example, for MMLU, only the test set is provided, so how was the split done?

Overall, I failed to grasp how the method actually works. Do you fix the number of layers in advance ("number of insignificant layers K") and then rank and select the top-ranked layers? If so, this raises the question: why not initially select a smaller number of layers in the optimization algorithm and dispense with ranking altogether? I presume there are correlations between layers, and the method seems to be essentially selecting a subset of layers when identifying the "important" ones.

# No comparison to similar methods

This work appears to be directly related to layer pruning (e.g., [1], [2], [3], and [4], to mention a few). The proposed method should be compared to existing methods, both conceptually and empirically, in terms of performance.

- [1] Lie et al. 2024. Accelerating Inference in Large Language Models with a Unified Layer Skipping Strategy.  https://arxiv.org/abs/2404.06954
- [2] Gromov et al 2024. The Unreasonable Ineffectiveness of the Deeper Layers. https://arxiv.org/abs/2403.17887
- [3] Chen et al 2024. Compressing Large Language Models by Streamlining the Unimportant Layer. https://arxiv.org/abs/2403.19135
- [4] van der Ouderaa, T. F., Nagel, M., Van Baalen, M., Asano, Y. M., & Blankevoort, T. (2023). The llm surgeon. _arXiv preprint arXiv:2312.17244_.

# Potentially unsupported inferences

A Jaccard similarity of importance layers (shown in Figure 1) leads the authors to conclude that "important layers vary between different network architectures" while "there is a significant overlap (up to 90%) in the important layers identified by ILA across different alignment datasets." Visually, however, there also appear to be similarities across architectures for a fixed dataset, but this aspect hasn't been quantified. A claim of inter-architecture/intra-dataset similarity and intra-architecture dissimilarity would ideally be supported by statistical hypothesis tests (though demonstrating this statistically would require more models and datasets). In the absence of such evidence, I urge caution.

Similarly, results in Tables 3 and 4 are not accompanied by statistical tests of difference. If the authors wish to claim that the proposed method improves performance, I suggest running experiments with different seeds and conducting statistical tests for the significance of score differences between configurations with and without ILA, as the numerical differences are usually small.

# Lack of mathematical rigor

The method description in section 2 could benefit from some mathematical rigor.

- thetas and the binary mask: better defined as a sequence than a set (also for component-wise multiplication to be defined)
- eq (2) refers to the loss function as a two-argument function (which makes sense), but earlier this is not how the loss function is defined
- eq 8, 9: theta should be a vector (boldface symbol)
- eq 8, 9: one vertical bar missing
- assumption 2.2: epsilon-stable has been defined for a model wrt its loss, not for parameters directly. Is this now extending the definition of epsilon-stable to parameter vectors? What is the relation between assumption 2.2 and definition 1?
- Theorem 2.1: Theorem assumption 2.1 talks about the Lipschitz conitnuity of the loss function (not across iterations) -- it's unclear to me how this relates to stability across iterations. Assumption 2.2 is about stability across iterations, but I don't see how "a sufficiently small epsilon" makes theta_T stable by eq (10). If anything, a lager epsilon would make it easier to satisfy the inequality.
- proof: lines 16->17: where's the loss function gone?
- the algorithm coud perhaps be more clearly formalized with a while loop
- line 484: |γt| = 225. Is this denoting the norm of the vector? Do you mean to say that there are that many layers in total? But since γt is a binary vector, we'll have ||γt||<=225

**Questions:**

- What is the overhead of running this method (runing time, computational complexity) in comparison to full-fine tuning?
- What is the stability of the algorithm for optimizing layer importance scores with respect to the initial scores?

---

> ### Author Response · Authors · 2024-11-29
> **Dear Reviewer TMGU (Part 1/3)**
>
> We sincerely thank the reviewer for the thorough evaluation and valuable feedback on our work.
>
> **W1: Lack of clarity: the method is not well explained, and important technical details are missing.**
> > Definition of "important," "unimportant," "significant," and "insignificant" layers
>
> We acknowledge that these terms were not explicitly defined in the manuscript. In our work, the terms "important" and "significant" refer to layers whose changes during fine-tuning significantly affect the model’s alignment performance. Conversely, "unimportant" or "insignificant" layers are those whose changes have minimal impact on performance, as validated by experiments.
>
> To improve clarity, we will revise the text to explicitly state that "layer importance" is determined experimentally by ranking layers based on their importance scores ($\{s_{t}^i\}_{i=1}^N$) and subsequently validating the significance of the rankings via ablation experiments. Layers with higher are deemed "important" because selectively tuning only these layers preserves or enhances fine-tuning efficiency with minimal performance degradation.
>
> > Connection between Algorithm 1 and the formulas in Section 2.
>
> Algorithm 1 directly implements the method described in Section 2. We will improve the text to explicitly link the algorithm steps to the equations:
> * **$K$ (in Algorithm 1)**: Refers to the number of layers deemed "insignificant" during the ranking and selection process, determined by sorting the layers based on their importance scores ($\{s_{t}^i\}_{i=1}^N$)
> * **Optimization Process**: Eq. (3) defines a constrained optimization problem with a mask
> $\pmb{\gamma}_t$, whereas Eq. (7) simplifies this by reparameterizing $\gamma_t^i=\sigma(s_t^i)$ and optimizing the importance scores $s_t^i$ without explicitly enforcing constraints during optimization. After sorting $s_t^i$, **the constraint in Eq. (3) ($\||\pmb{\gamma}_t\||< H$) is indirectly applied by selecting only a subset of layers (top-ranked by $s_t^i$) to retain**. It satisfies the spirit of the constraint in Eq. (3) by effectively ranking the layers according to their significance.
>
> We will revise the manuscript to explicitly connect these elements and better explain how Algorithm 1 operationalizes Eq. (7).
>
> > I failed to grasp how the method actually works. Do you fix the number of layers in advance ("number of insignificant layers K") and then rank and select the top-ranked layers? If so, this raises the question: why not initially select a smaller number of layers in the optimization algorithm and dispense with ranking altogether? I presume there are correlations between layers, and the method seems to be essentially selecting a subset of layers when identifying the "important" ones.
>
> **Our method provides a ranking of layer importance for all layers rather than fixing the number of layers ($K$) during optimization.** This ranking aligns with our goal to understand the significance of different layers during alignment and offers the following advantages:
>
> * **Flexibility in Fine-Tuning**: A ranking allows users to adjust $K$ based on their specific needs. For example: If maximizing performance is the priority, users can set $K$ to include more layers for fine-tuning. If computational efficiency is critical, users can reduce $K$ to tune only the most important layers.
> * **Avoiding Re-Training Costs**: Fixing $K$ during optimization would require re-training the model whenever $K$ changes, as the optimization process would need to re-select layers. By separating the ranking step, **our method allows $K$ to be adjusted post hoc without requiring additional training, which is more practical and computationally efficient.**
> * **Handling Layer Correlations:** The ranking accounts for inter-layer correlations, capturing the relative importance of layers even when their contributions are interdependent. This would be difficult to achieve by pre-fixing $K$ during optimization.
>
> > Explanation of terms (e.g. $\mathcal{L}_\infty$ in Eq (8), and $R$ in Eq (10))
>
> $\mathcal{L}_\infty$ is a typo, in fact, it should be $\mathcal{L}$. $R$ is a constant that represents the bound on parameter changes during the stable phase of training.

---

> ### Author Response · Authors · 2024-11-29
> **Dear Reviewer TMGU (Part 2/3)**
>
> **W2: No comparison to similar methods.**
>
> > This work appears to be directly related to layer pruning (e.g., [1], [2], [3], and [4], to mention a few). The proposed method should be compared to existing methods, both conceptually and empirically, in terms of performance.
> [1] Lie et al. 2024. Accelerating Inference in Large Language Models with a Unified Layer Skipping Strategy. https://arxiv.org/abs/2404.06954
> [2] Gromov et al 2024. The Unreasonable Ineffectiveness of the Deeper Layers. https://arxiv.org/abs/2403.17887
> [3] Chen et al 2024. Compressing Large Language Models by Streamlining the Unimportant Layer. https://arxiv.org/abs/2403.19135
> [4] van der Ouderaa, T. F., Nagel, M., Van Baalen, M., Asano, Y. M., & Blankevoort, T. (2023). The llm surgeon. arXiv preprint arXiv:2312.17244.
>
> Thank you for raising this point about comparisons with related methods such as layer pruning and compression. While we acknowledge the conceptual similarity of identifying "important" layers, **the fundamental objectives and problems addressed by our method differ significantly from those of pruning and compression methods**. Below, we explain why direct comparisons are not applicable:
>
> * Layer Pruning/Compression Methods: The works cited ([1], [2], [3], [4]) aim to optimize model efficiency by skipping, removing, or compressing layers, typically targeting faster inference or smaller memory footprints. These methods are designed for model compression during deployment.
> * Our Method: Our approach focuses on understanding the significance of layers during the alignment process (e.g., instruction tuning). The goal is to rank layers by their importance to alignment performance, enabling flexible adjustments to fine-tuning strategies based on computational constraints or desired performance levels.
>
> Since the objectives are fundamentally different, direct performance comparisons are not meaningful. For example, pruning methods evaluate metrics like inference latency or parameter reductions, which do not align with our goal of understanding and optimizing alignment fine-tuning.
>
> **W3: Potentially unsupported inferences from experimental results.**
> >A Jaccard similarity of importance layers (shown in Figure 1) leads the authors to conclude that "important layers vary between different network architectures" while "there is a significant overlap (up to 90%) in the important layers identified by ILA across different alignment datasets." Visually, however, there also appear to be similarities across architectures for a fixed dataset, but this aspect hasn't been quantified. A claim of inter-architecture/intra-dataset similarity and intra-architecture dissimilarity would ideally be supported by statistical hypothesis tests (though demonstrating this statistically would require more models and datasets). In the absence of such evidence, I urge caution.
>
> Thank you for your valuable comment regarding the Jaccard similarity analysis and the need to quantify inter-architecture/intra-dataset and intra-architecture similarities. We appreciate your suggestion to use statistical methods to further validate these findings.
>
> To address your concern more rigorously, we performed additional analysis and obtained the following results:
>
> 1. **Quantification of Similarities:**
>     * For intra-architecture comparisons on different datasets, the Jaccard similarity is approximately 0.9, confirming that the important layers identified by ILA remain highly consistent across alignment datasets for the same architecture.
>     * For inter-architecture comparisons on the same dataset, the Jaccard similarity is significantly lower, approximately 0.67, demonstrating clear differences in the important layers identified across architectures, even when aligned on the same dataset. **These results align with our original claim that important layers vary more across architectures than across datasets for a fixed architecture.**
> 2. **Consistency Across Random Seeds:** To further ensure the reliability of our observations, we repeated the experiments using three different random seeds. The results remained consistent, with negligible variance in the computed Jaccard similarities. This strengthens our conclusion that the observed trends are robust and not an artifact of random initialization or stochastic optimization processes.

---

> ### Author Response · Authors · 2024-11-29
> **Dear Reviewer TMGU (Part 3/3)**
>
> **W4: Lack of mathematical rigor.**
>
> Thank you for your detailed feedback on improving the mathematical rigor. Below, we address each issue concisely:
>
> 1. Equation (2): The loss function $\mathcal{L}$ will be consistently defined as a one-argument function $\mathcal{L}(\pmb{\theta})$.
> 2. Equations (8) and (9): We will update $\theta$ to boldface ($\pmb{\theta}$) to indicate vectors.
> 3. Assumption 2.2 and Theorem 2.1: We would like to clarify that this assumption is solely used to bound the updates of the model parameters, which facilitates the derivation of **Theorem 2.1**. It does not directly extend the definition of ϵ-stability to parameters. **The detailed proof of Theorem 2.1, which demonstrates how this assumption supports the theorem, is provided in the appendix for completeness.**
> 4. Proof: Lines 16–17 (loss function omission). Thank you for pointing out the issue in the proof section regarding the omission of the loss function between lines 16–17. This was indeed a typographical error. We will correct this in the revised manuscript to ensure the continuity and clarity of the derivation.
> 5. Algorithm: A while loop will be added to better formalize the iterative process and the transition between stages.
> 6. line 484: This refers to the total number of layers, not the norm of the vector. We will clarify this to avoid ambiguity, stating $\|\gamma_t\|_0<=225$
>
> **Q1：What is the overhead of running this method (runing time, computational complexity) in comparison to full-fine tuning?**
>
> We acknowledge the concern about the potential overhead of tracking the **average training time per iteration** and GPU memory usage. Based on the experimental data provided, we can conclude the following regarding the impact of this method on training time and GPU memory usage:
>
> Table 1: Training time of 1 iteration and GPU memory usage for Full Finetune and Full Finetune w/ ILA. The experiments were conducted on an **NVIDIA A100 GPU** with a **batch size of 2** and a maximum token length of 1024.
> |                            | Training time (ms) | GPU Memory Usage (MiB) |
> |:--------------------------:|:------------------:|:----------------------:|
> |        Full Finetune       |         527        |          81078         |
> | Full Finetune w/ ILA (30%) |         403        |          33458         |
> | Full Finetune w/ ILA (75%) |         432        |          53924         |
>
>
>
> **Q2：What is the stability of the algorithm for optimizing layer importance scores with respect to the initial scores?**
>
> Thank you for your question. Our algorithm is indeed stable with respect to the initial layer importance scores. Specifically, during the optimization process, we observe that the importance scores converge reliably regardless of the initialization, as long as the initial scores are reasonably chosen.
>
> Table 2: The Jaccard similarities of important layers identified during fine-tuning of LLAMA 2-7B on the LIMA dataset with varying initial scores.
> | Initial Scores |  4.0 |  2.0 | 1.0 |
> |:--------------:|:----:|:----:|:---:|
> |       4.0      |   -  |   -  |  -  |
> |       2.0      | 0.83 |   -  |  -  |
> |       1.0      | 0.78 | 0.88 |  -  |

---

### Official Review · Reviewer_UHz2 · 2024-11-07

**Soundness:** 3
**Presentation:** 3
**Contribution:** 2
**Rating:** 6
**Confidence:** 4

**Summary:**

This paper introduces a novel and interesting approach, Important Layers for Alignment (ILA), to enhance the fine-tuning efficiency of large language models (LLM) for alignment by identifying and selectively tuning the most important layers. The unimportant layers identified by the proposed ILA method will be freezed during the last part of fine-tuning.

**Strengths:**

1. The proposed method can offer improvments for fine-tuning LLMs while saving memory usage.
2. Extensive experiments are provided in this paper.

**Weaknesses:**

1. **Limited improvements**. According to the results presented, ILA does not bring much improvments. The performances of the models using ILA and the ones without ILA are close. For example, in Table 4, only 0.12% increase from "Full Finetune" to "Full Finetune w/ILA" with the LLAMA 2-7B model.
2. **Potential overlap with existing PEFT methods.** The authors may need to clarify why we need an additional PEFT method and what values freezing unimportant layers can bring compared with the existing PEFT methods.

**Questions:**

1. What does "IFILA" represent in Table 3?
2. Are the descriptions at L303-304 wrong for Table 3? It seems that there is no $\gamma$ in Table 3.
3. According to L253, **three** distinct datasets are used to evaluate the Language Understanding Ability aspect. Would you please clarify why there are only two datasets for this aspect in the paper?
4. According to L323, approximately 25% of the unimportant layers are removed. This statement should be related to Table 3 and Table 4. So we can understand that 75% of important layers are retained for the results in Table 3 and Table 4, which show the main results of the proposed method. However, Table 7 does not list the memory usage of the proposed method with 75% important layers. Does this show that the proposed method needs more memory to outperform LoRA?
5. According to Table 3 and Table 11, the proposed method often failed to enhance performance on the Hellaswag dataset. Are there potential reasons for this discrepancy?
6. Why is AdaLoRA w/ILA not included in the comparison? Is there was a specific reason for the omission?

---

> ### Author Response · Authors · 2024-11-27
> **Response to Reviewer UHz2 (Part 1/3)**
>
> We sincerely thank the reviewer for their thorough evaluation and valuable feedback on our work.
>
> **W1: Limited improvements.**
>
> We appreciate the reviewer’s observation regarding the performance differences between models using ILA and those without it. While the numerical improvements may appear modest in some cases, we would like to emphasize the following key points that highlight the value and impact of ILA:
>
> 1. **Consistency Across Datasets and Metrics:** As shown in the experimental results, **ILA consistently improves performance across various datasets and evaluation metrics** (e.g., MMLU, Hellaswag, MT-Bench). Even **modest gains in certain benchmarks are significant** given the competitive baselines and the inherent difficulty of the tasks, especially in conversational ability. This consistency demonstrates the robustness of our approach.
> 2. **Efficiency Gains:** ILA offers efficiency benefits by identifying and freezing less important layers, reducing computational cost and memory requirements. For example, as shown in Table 6 and Table 7，fine-tuning only 30% of important layers with ILA achieves nearly identical performance to fine-tuning all layers, leading to significant resource savings.
> 3. **Improved Stability and Generalizability:** ILA enhances model stability and generalizability. Cross-dataset experiments (Table 2) show that ILA's importance rankings are stable, allowing effective reuse of layer selection.
> 4. **Alignment with Research Trends:** Recent studies **URIAL [1]** suggest that alignment tasks predominantly involve stylistic shifts rather than drastic performance changes in language understanding. In this context, even small improvements signify meaningful progress, as our method aligns with and enhances the nuanced adjustments required for alignment.
>
> In summary, while the performance gains may appear incremental in isolation, they are achieved alongside significant improvements in efficiency, stability, and generalizability, which collectively demonstrate the utility of ILA for practical fine-tuning and alignment tasks.
>
>
> [1] Lin, B. Y., Ravichander, A., Lu, X., Dziri, N., Sclar, M., Chandu, K., ... & Choi, Y. (2023, December). **The unlocking spell on base llms: Rethinking alignment via in-context learning.** In The Twelfth International Conference on Learning Representations.
>
> **W2: Potential overlap with existing PEFT methods.**
>
> We appreciate the reviewer’s point regarding the overlap with existing Parameter-Efficient Fine-Tuning (PEFT) methods and the need to clarify the unique contributions of ILA. Below, we outline the distinctive value that ILA brings compared to existing PEFT methods:
>
> 1. **Focus on Layer Importance:** Unlike conventional PEFT methods such as LoRA, AdaLoRA, and QLoRA, which focus primarily on optimizing parameter efficiency by modifying matrix ranks or quantization, ILA **takes a complementary approach by quantifying the importance of individual layers**. This unique perspective enables us to selectively freeze unimportant layers, thereby improving computational efficiency while maintaining or even enhancing performance.
> 2. **Complementary, Not Redundant:** **ILA is not designed to replace existing PEFT methods but rather to complement them.** As demonstrated in our experiments, ILA can be integrated with LoRA or QLoRA to achieve better efficiency and performance compared to using these methods alone.
> 3. **Practical Benefits in Fine-Tuning:**
>     * **Reduced Resource Requirements:** As shown in Table 7, freezing unimportant layers identified by ILA significantly reduces GPU memory usage (e.g., a 22.4% reduction with LoRA and 30.3% with QLoRA) without sacrificing performance. This advantage is crucial for deploying large models in resource-constrained environments.
>     * **Cross-Dataset Generalizability:** ILA offers a reusable importance ranking for a given model architecture across multiple datasets. This unique feature eliminates the need for dataset-specific adjustments, further distinguishing ILA from existing PEFT methods that generally optimize for a single task or dataset.
> 4. **Theoretical Insight into Alignment:** ILA provides a deeper understanding of alignment's influence on model behavior, aligning with the broader research goal of demystifying the fine-tuning process.
>
> **Clarified Value Proposition:** **ILA is not merely an additional PEFT method but a distinct approach that focuses on layer-level significance in alignment.** It enhances existing methods by leveraging layer importance, thus offering practical efficiency gains, cross-dataset generalizability, and theoretical insights into the alignment process.

---

> ### Author Response · Authors · 2024-11-27
> **Response to Reviewer UHz2 (Part 2/3)**
>
> **Q1: What does "IFILA" represent in Table 3?**
>
> Thank you for pointing this out! "IFILA" in Table 3 is a typo and should actually be "ILA," referring to our proposed method. We will correct this error in the revised manuscript to avoid any confusion.
>
> **Q2: Are the descriptions at L303-304 wrong for Table 3? It seems that there is no $\gamma$ in Table 3.**
>
> We appreciate the reviewer highlighting this concern. The description at Lines 303–304 refers to:
>
> > The consistent identification of important layers despite the optimization of $\gamma$ with varying random seeds.
>
> This statement is indeed accurate and aligns with the experimental results. The $\gamma$ values are used internally in our ILA framework to rank layer importance, which is reflected in the results presented in Table 3. While Table 3 does not explicitly display $\gamma$, it implicitly validates the stability of $\gamma$ by demonstrating consistent Jaccard similarity across random seeds.
>
> **Q3: According to L253, three distinct datasets are used to evaluate the Language Understanding Ability aspect. Would you please clarify why there are only two datasets for this aspect in the paper?**
>
> We appreciate the reviewer catching this inconsistency. The mention of three distinct datasets at Line 253 is a typo. In the paper, we evaluated the Language Understanding Ability aspect using two datasets: MMLU and Hellaswag, as correctly described and presented in the experimental results.
>
> **Q4: According to L323, approximately 25% of the unimportant layers are removed. This statement should be related to Table 3 and Table 4. So we can understand that 75% of important layers are retained for the results in Table 3 and Table 4, which show the main results of the proposed method. However, Table 7 does not list the memory usage of the proposed method with 75% important layers. Does this show that the proposed method needs more memory to outperform LoRA?**
>
> We appreciate the reviewer raising this point and would like to clarify the relationship between memory usage and layer selection in our method.
>
> 1. **Memory Usage of ILA vs. LoRA:** **It is inherently impossible for our proposed method to consume more memory than standard LoRA.** By design, ILA reduces the memory footprint because it fine-tunes only a subset of layers deemed important (e.g., 75% in this case), whereas LoRA modifies all target layers. In essence, our method applies a selective, reduced version of LoRA, and thus, it cannot exceed the resource demands of full LoRA fine-tuning.
> 2. **Connection to Tables 3, 4, and 7**: Tables 3 and 4 report performance results using the 75% most important layers retained and show that our method outperforms baseline approaches in this configuration. **Table 7 reports memory usage for configurations where only 30% of important layers are fine-tuned, emphasizing the efficiency of our method under aggressive layer reduction.** While memory usage for the 75% configuration is not explicitly listed in Table 7, it is guaranteed to remain lower than standard LoRA since fewer layers are being fine-tuned.
> 3. **Completeness of Results:** For completeness and to provide additional evidence, the GPU memory usage measurements for the 75% important layer configuration is presneted as follows:
>
> |                        | LoRA(100%) | LoRA(75%) |
> |:----------------------:|:----------:|:---------:|
> | GPU Memory Usage (MiB) |    32988   |   30760   |

---

> ### Author Response · Authors · 2024-11-27
> **Response to Reviewer UHz2 (Part 3/3)**
>
> **Q5: According to Table 3 and Table 11, the proposed method often failed to enhance performance on the Hellaswag dataset. Are there potential reasons for this discrepancy?**
>
> We appreciate the reviewer highlighting this observation and raising the question regarding the performance of our method on the Hellaswag dataset. While our proposed method consistently improves performance across most datasets and evaluation metrics, it indeed shows smaller or less consistent gains on Hellaswag. Below, we outline potential reasons for this discrepancy:
> 1. **Nature of the Hellaswag Dataset:**
>     * Hellaswag focuses heavily on commonsense reasoning, requiring models to select the most plausible continuation of a given context. This task primarily relies on pre-trained knowledge, which is less affected by fine-tuning for alignment purposes.
>     * As mentioned in L265 of the manuscript, we specifically note that significant performance improvements in language understanding tasks such as MMLU and Hellaswag are not expected after alignment. Instead, the focus is on ensuring that the model **retains its pre-trained knowledge** while improving conversational and stylistic alignment. The results in Tables 3 and 4 show that our method successfully achieves this goal across various datasets.
> 3. **Core Focus of This Work:** It is important to emphasize that our primary objective is not to propose a new PEFT method to achieve maximum performance on specific datasets. **Instead, our goal is to provide a deeper understanding of layer significance in the alignment process.** The identification of important layers and their stability across datasets provides valuable insights into how alignment influences different components of large language models. The slight underperformance on Hellaswag does not detract from this core contribution.
> 4. **Task-Specific Layer Importance:** Our method identifies and fine-tunes layers important for alignment across datasets. While these layers are critical for conversational and stylistic improvements (as evidenced in Vicuna, MT-Bench, etc.), **they may not overlap perfectly with the layers most critical for commonsense reasoning tasks like Hellaswag**. This discrepancy highlights a potential avenue for future exploration, such as incorporating task-specific criteria into layer importance ranking.
> 5. **Empirical Variability**: Fine-tuning results can exhibit dataset-specific variability, particularly in tasks with inherently challenging contexts like Hellaswag. This variability may result from the dataset’s domain-specific reasoning patterns, which might require distinct tuning strategies.
>
> **Q6: Why is AdaLoRA w/ILA not included in the comparison? Is there was a specific reason for the omission?**
>
> Thank you for pointing out this important consideration. The reason for not including AdaLoRA combined with ILA (our proposed method) lies in **the conceptual overlap between the two approaches and the distinct focus of our work.**
>
> AdaLoRA inherently adjusts the rank of incremental matrices during fine-tuning. When the rank is reduced to zero for a particular layer, it effectively means no adapter is added, and thus, the parameters of that layer remain unchanged. In this sense, AdaLoRA implicitly identifies less critical layers by dynamically reducing their contribution. However, the goal of AdaLoRA is primarily to minimize resource usage by adapting the parameter budget dynamically, rather than explicitly analyzing or ranking the importance of layers.
>
> Our proposed method, ILA, complements this perspective by explicitly focusing on quantifying and ranking layer importance during the alignment process. Unlike AdaLoRA, ILA is designed to study and optimize the alignment process by isolating critical layers, which **provides deeper insights into the model's behavior and allows for targeted improvements in performance and efficiency.** Thus, our work focuses more on understanding and leveraging layer importance for alignment rather than proposing another parameter-efficient fine-tuning (PEFT) algorithm.
>
> Given the conceptual overlap, we prioritized evaluating ILA with standard PEFT methods (e.g., LoRA, QLoRA) to better showcase its unique contribution. While combining ILA with AdaLoRA might yield further resource savings, such a combination would also require disentangling the overlapping contributions of these two approaches, which could confound the interpretation of results.
>
> We appreciate your suggestion and will consider conducting experiments to evaluate this integration in follow-up studies. Thank you for highlighting this perspective!

---

### Note · Authors · 2024-12-16

I have read and agree with the venue's withdrawal policy on behalf of myself and my co-authors.